



# Hydrological connectivity controls dissolved organic carbon exports in a peatland-dominated boreal catchment stream

Antonin Prijac[1,2,3], Laure Gandois[4], Pierre Taillardat[1,2,5], Marc-André Bourgault[6], Khawla Riahi[7], Alex Ponçot[1], Alain Tremblay[8], Michelle Garneau[1,2,3]

[1] Centre de Recherche sur la Dynamique du Système Terre (GÉOTOP), Université du Québec à Montréal, Canada
[2] Groupe de Recherche Inter-universitaire en Limnologie (GRIL), Université du Québec à Montréal, Canada
[3] Institut des Sciences de l'Environnement (ISE), Université du Québec à Montréal, Canada
[4] Laboratoire Écologie Fonctionnelle et Environnement, UMR 5245, CNRS-UPS-INPT, Toulouse, France
[5] NUS Environmental Research Institute, National University of Singapore, Singapore
[6] Département de Géographie, Université Laval, Québec, Canada
[7] Centre Eau, Terre et Environnement, Institut National de la Recherche Scientifique, Québec, Canada
[8] Programme Gaz à Effet de Serre, Hydro Québec, Montréal, Québec, Canada

*Correspondence to*: Antonin Prijac (antonin.prijac@gmail.com) and Laure Gandois (laure.gandois@cnrs.fr)

**Abstract**

Peatland-derived dissolved organic carbon (DOC) exports from boreal peatlands are variable during the ice-free season, depending on the peatland water table and the alternation of low and high flow in peat-draining streams. However, calculation of the specific DOC exports from a peatland can be challenging considering the multiple potential DOC sources within the catchment. A calculation approach based on the hydrological connectivity between the peat and the stream could help to solve this issue, an approach used in the present study. This study took place from June 2018 to October 2019 in a boreal catchment in north-eastern Canada, with 76.7% of the catchment covered by ombrotrophic peatland. The objectives were (1) to establish relationships between DOC exports from a headwater stream and the peatland hydrology; (2) to quantify, at the catchment scale, the amount of DOC laterally exported to the draining stream; and (3) to define the patterns of DOC mobilization during high river flow events. At the peatland headwater stream outlet, the DOC concentrations were monitored at a high frequency (hourly) using a fluorescent dissolved organic matter (fDOM) sensor, a proxy for DOC concentrations. Hydrological variables, such as stream outlet discharge and the peatland water table depth (WTD), were continuously monitored for 2 years. Our results highlight the direct and delayed control of subsurface flow from peat to the stream and associated DOC exports. Rain events raised the peatland WTD, which increased the hydrological connectivity between the peatland and the stream. This led to increased stream discharge ($Q$) and a delayed DOC concentration increase, typical of lateral subsurface flow. The magnitude of the WTD increase played a crucial role in influencing the quantity of exported DOC. Based on the assumption that the peatland is the major contributor to DOC exports and other DOC sources were negligible during high-flow periods, we propose a new approach to calculate the specific DOC exports attributable to the peatland by distinguishing the surface used to the calculation between high-flow and low-flow periods. In 2018–2019, 92.6% of DOC was exported during flood events, despite accounting for 59.1% of the period. In 2019–2020, 93.8% of DOC was exported during flood events, which represented 44.1%





of the period. Our analysis of individual flood events revealed three types of events and DOC mobilization patterns. The first type is characterized by high rainfall leading to an important WTD increase favouring the connection between the peatland and the stream, leading to high DOC exports. The second is characterized by a large WTD increase succeeding a previous event that had depleted DOC available to be transferred to the stream, leading to lower DOC exports. The third type corresponds to low rainfall events with an insufficient WTD increase to reconnect the peatland and the stream, leading to low DOC exports. Hence, DOC exports are sensitive to hydroclimatic conditions. Moreover, flood events, changes in rainfall regimes, the ice-free season duration and porewater temperatures may affect the exported DOC and, consequently, partially offset the carbon sequestration capacity of peatlands.

## 1. Introduction

At the global scale, boreal peatlands are the main contributors of dissolved organic carbon (DOC) exported to the aquatic continuum, accounting for 58% of the global exports (Rosset et al., 2022). In the context of a net ecosystem carbon budget, quantifying DOC exports, as well as particulate organic carbon (POC) and dissolved inorganic carbon (DIC) exports, is crucial to evaluate how much carbon is lost through this pathway (Webb et al., 2019). Ignoring those carbon losses may, in some cases, overestimate annual peatland carbon accumulation by 40%–80% (Roulet et al., 2007). DOC is the main form of exported carbon and accounts for 54.3%–91% of the total aquatic carbon exported (Roulet et al., 2007; Worrall et al., 2009; Holden et al., 2012; Dinsmore et al., 2013; Leach et al., 2016). Moreover, DOC can be mineralized along the aquatic continuum and get converted into dissolved $CO_2$ (Aho and Raymond, 2019). Hence, lateral DOC exports from peatland headwater streams are important to quantify considering it can lead to greenhouse gas (GHG) emissions to the atmosphere (Billett et al., 2012; Wallin et al., 2013; Rasilo et al., 2017).

One challenge related to net ecosystem carbon budget assessment is that, within a catchment, DOC export to stream(s) comes from the different ecosystems (i.e. forest, wetlands, etc.) in the landscape (Webb et al., 2018). Thus, it is methodologically challenging to differentiate the respective contributions of each ecosystem (Billett et al., 2006, 2012; Tipping et al., 2010; Rosset et al., 2019). However, peatlands are recognized as hotspots for production and transfer of DOC through lateral discharge (including subsurface runoff and porewater seepage) to stream networks (Freeman et al., 2001; Laudon et al., 2011; Rosset et al., 2019; Zhu et al., 2022). Strong and positive correlations have already been established between the surface of a catchment covered by peat and the exported DOC to surface waters (Billett et al., 2006; Laudon et al., 2011; Olefeldt et al., 2013).

To obtain a precise estimate of the peatland contribution in DOC exports, a specific DOC export (i.e. a flux normalized to a surface) that includes the peatland surface area within the catchment must be determined. Most of the previous studies have presented DOC exports normalized to the surface of peatland-dominated catchments rather than normalized to the peatland surface area within the catchment (Köhler et al., 2008, 2009; Worrall et al., 2009; Dinsmore et al., 2013; Dick et al., 2015), possibly leading to underestimation of DOC exports. Leach et al. (2016) proposed calculating the specific exports using



both total catchment area and peatland surfaces in the catchment as a way to report minimum and maximum values of DOC exports. The minimum value of the specific exports uses the catchment area as a reference, based on the hypothesis that DOC exported from the peatland is equivalent to DOC exported from the non-peatland areas. The maximum value of the specific exports is calculated by using the peatland area and considered that the DOC contribution from non-peatland ecosystems can be negligible. Another approach to obtain peatland-specific DOC exports is by subtracting the sum of DOC entering the peatland to DOC exports at the peatland outlet (Rosset et al. 2019). Unfortunately, this approach is not scalable to all peatlands given the variability in catchment configurations.

Recent advances in high-frequency measurements of fluorescence of dissolved organic matter (fDOM), a quantitative proxy of DOC, has allowed researchers to accurately measure DOC exported at high temporal frequency (Tunaley et al., 2016; Rosset et al., 2019; Blaurock et al., 2021). This high-frequency monitoring is essential to catch DOC export variations during flood events, which are believed to be crucial moments of DOC transfers (Tipping et al., 2010; Raymond et al., 2016). Pulses of DOC during flood events can be understood as a succession of hydrological connection and disconnection between the peatland and the stream, causing changes in DOC concentrations in the stream (Billett et al., 2006; Laudon et al., 2011; Jutebring Sterte et al., 2022). The runoff generation into the peat is controlled by the water table depth (WTD; Holden and Burt, 2003; Frei et al., 2010), where a large WTD increase during flood events leads to hydrological reconnection between DOC sources (Inamdar et al., 2004; Tunaley et al., 2016; Rosset et al., 2020) and greater DOC exports (Blaurock et al., 2021).

Advances in high-frequency monitoring and better effort directed towards flood events have confirmed that the majority of DOC is exported from peatlands during flood periods rather than during recession periods (Dick et al., 2015; Birkel et al., 2017; Blaurock et al., 2021). During flood events, DOC exports in the catchment dominated by peatlands are mainly composed of recently produced carbon derived from peat (Tipping et al., 2010; Billett et al., 2012; Holden et al., 2012; Juutinen et al., 2013; Dean et al., 2019). Recent studies have pointed out the importance of characterizing DOC export variability rather than identifying their sources to understand the processes underlying DOC mobilization (Birkel et al., 2017; Blaurock et al., 2021; Zhu et al., 2022).

DOC exports during flood events may vary depending on many parameters: the magnitude of the rainfall events, the season and the porewater temperature, the recurrence of high-flow events, the presence of a free-rainfall period, and the antecedent wetness of the catchment (Leach et al., 2016; Tiwari et al., 2018; Rosset et al., 2020; Blaurock et al., 2021). Previous studies have highlighted that long periods between rainfall events favour DOC production. Greater DOC exports are measured once the hydrological connection is restored, given the large amounts of DOC recently produced in the peatland and which could be mobilized through lateral discharge (Worrall et al., 2008; Clark et al., 2009; Grand-Clement et al., 2014; Buzek et al., 2019). Others have shown that great WTD before a rain event favour rapid DOC mobilization and lead to greater exports, independently of the recurrence between events in a peatland (Birkel et al., 2017; Blaurock et al., 2021). The amount of exported DOC is also controlled by production processes, stimulated by the temperature (Clark et al., 2007, 2009; Grand-Clement et al., 2014; Zhu et al., 2022) because DOC concentrations in the peat pore water increase with the temperature (Freeman et al., 2001; Buzek et al., 2019).





Among the studies that have used an event-based approach in peatland streams, most of them have been performed in temperate (Worrall et al., 2008; Austnes et al., 2010; Grand-Clement et al., 2014; Tunaley et al., 2016) and alpine (Birkel et al., 2017; Rosset et al., 2020) catchments, and none have been realized in boreal environments. Boreal catchments are constrained by seasonal freezing and pronounced snowmelt (Ågren et al., 2010; Leach et al., 2016; Tiwari et al., 2018) that potentially affect and delay DOC exports (Laudon et al., 2012). The seasonal and interannual variability also influence DOC
production.

    Considering the particular climatic context of boreal peatlands and the importance of hydrological processes on DOC exports, this study aimed to characterize patterns of DOC exports from a boreal peatland headwater stream over two consecutive years. Based on high-frequency DOC concentrations and different hydrological parameters including rainfall, stream discharge and WTD, we used an event-based approach to document the mechanisms driving DOC mobilization and
exports during flood events. Individual flood events were compared in order to understand how hydrological and meteorological variables control the amount of exported DOC. This study comprises three research objectives: (1) to establish relationships between DOC exports from a headwater stream and the peatland hydrology; (2) to quantify, at the catchment scale, the amount of DOC laterally exported to the draining stream; and (3) to define the patterns of DOC mobilization during high river flow events.

**2. Study site**

The study site, located in north-eastern Canada within the Romaine River catchment (14 500 km$^2$), adjacent to the Labrador border, was previously described in Prijac et al. (2022). It is located in the eastern spruce–moss bioclimatic domain of the closed boreal forest (Payette, 2001) at the limit of the coastal plain and the Highlands of the Laurentian Plateau of the Precambrian Shield (Dubois, 1980). The Bouleau peatland study site (50°31'N, 63°12'W; altitude 108 ± 5 m) is an
ombrotrophic, slightly dome-shaped bog positioned at the head of a catchment (Fig. 1). Peat accumulation was initiated ca. 9260 calibrated years before present, and the maximum peat depth reaches 440 cm (Primeau and Garneau, 2021). The surface microtopography of the peatland shows a clear patterned surface of alternating dry hummocks, lawns, hollows and pools. The surface vegetation varies according to the microtopography, with *Sphagnum fuscum*, *S. capillifoium* and *Cladonia rangiferina* on the hummocks; *S. magellanicum*, *S. rubellum*, *S. cuspidatum* and *Trichophorum cespitosum* on the lawns; and *Sphagnum*
*majus* and *S. pulchrum* on the hollows (Primeau and Garneau, 2021).

    The study focused on the outlet of the peatland drained by a headwater stream of about 3 km in length, which flows north to south across the peatland from the western side. The surface of the catchment drained by the stream is 2.22 km$^2$ and the area of the catchment covered by peat is 1.70 km$^2$.



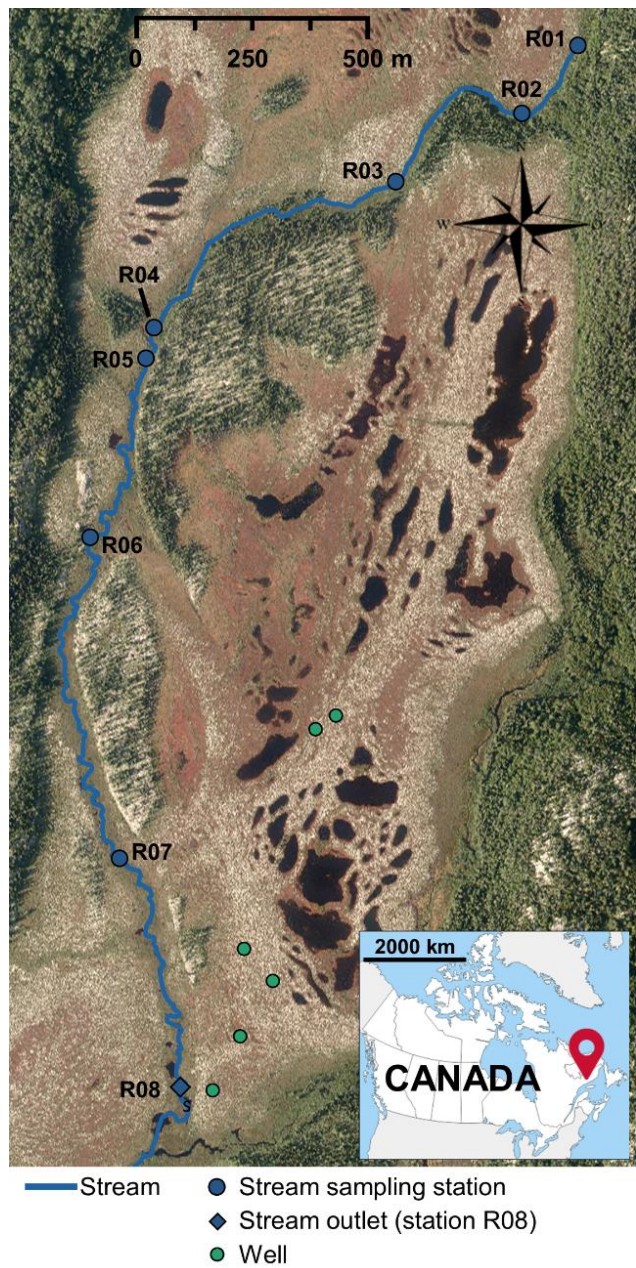

**Figure 1.** Aerial photo of the Bouleau peatland with the location of the wells where water-level data loggers have been installed (green dots), sampling sites along the stream (blue dots) and the outlet of the peatland drainage stream (the aerial photo is provided by Hydro-Québec).

As described in Prijac et al. (2022), based on the regional climate data, the mean annual temperature is 1.5°C and the total annual precipitation is 1011 mm, of which 590 mm falls as snow. An average monthly positive temperature occurs from





May to October with 1915 growing degree days above zero (Havre-Saint-Pierre meteorological station, mean 1990–2019, Environment of Canada). During the growing season, the average air temperature was $13.2 \pm 6.9°C$, with a minimum of $-7.9°C$ in early October 2018 and a maximum of $30.8°C$ in late July 2018. The warmest month was July 2018, with an average monthly temperature of $17.9 \pm 5.6°C$, and the coldest month was October 2018, with an average monthly temperature of $3.52 \pm 5.29°C$. Average rainfall events were 7 mm day$^{-1}$ and maximum daily rainfall was in July 2018, with 41 mm day$^{-1}$. The wettest month was August 2019, with total rainfall of 129 mm, while the driest month was July 2019, with total rainfall of 27 mm.

The measurement period started in June 2018. Consequently, meteorological, hydrological and physicochemical variables are presented for the growing season defined from June to October, as described by Prijac et al. (2022). Annual DOC exports are presented for two complete periods of 12 months ranging from June 2018 to May 2019 for the first year and from June 2019 to May 2020 for the second year.

## 3. Methods

### 3.1 Water sampling

Manual water sampling along the studied headwater stream was performed during the same sampling periods as described in Prijac et al. (2022): five times during the 2018 growing season (14 June, 12 July, 8 August, 1 September and 10 October) and four times in 2019 (8 June, 3 August, 5 September and 10 October).

Stream surface water was collected at 11 sampling stations along the headwater stream (Fig 1). Samples were also collected from three tributaries at about 10 m before the confluence (Fig. 1). Because the stream was intermittent in the upstream section during the growing season, stations R01 and R02 were not sampled during each campaign.

The physicochemical parameters (temperature, pH, specific conductivity and dissolved oxygen saturation) were measured from a multi-parameter portable meter (Multiline Multi-3620 IDS, WTW, Germany) at each sampling site. All water samples were collected in polypropylene bottles previously cleaned with ultra-pure water and rinsed with sample water. The samples were filtered on GF/F filters (Whatman) that had been pre-combusted for 4 h at 450°C.

### 3.1.1 Analyses of DOC concentrations

Filtered water was prepared for DOC analysis, following the method described in Prijac et al. (2022), by acidification to pH 2 with 1 M HCl and stored in 40 mL glass vials. DOC and total nitrogen (TN) concentrations were analysed using the catalytic oxidation method followed by non-dispersive infrared (NDIR) detection of produced $CO_2$ (TOC analyser TOC-L, Shimadzu, Japan) with a limit of quantification of 0.1 mg C L$^{-1}$.



### 3.2 *In situ* high-frequency monitoring

#### 3.2.1 fDOM and physicochemical parameters

An EXO2 multi-parameter probe (YSI, USA) was placed at the stream outlet, at the same station where discharge was monitored and approximately 40 cm above the stream bed. The physicochemical parameters (water temperature, pH, specific conductivity and dissolved oxygen concentration and saturation) were recorded hourly from June 2018 to May 2020 and calibrated about once a month during the growing season.

The parameters monitored included the fluorescence of DOM (fDOM) measurements ($\lambda_{excitation} = 365 \pm 5$ nm / $\lambda_{emission} = 480 \pm 40$ nm) and turbidity. The time series includes the removal of some fDOM measurements when the probe was found in the stream sediments from mid-July to mid-August 2018 and in July and late August 2019. fDOM measurements were removed when turbidity exceeded a threshold of 50 FNU as it might alter the values (de Oliveira et al., 2018). Except for the periods when the probe was found in the sediments, there was no important turbidity peak, so the study focused on DOC.

A total of 826 individual measurements were removed in 2018, corresponding to 26.2% of data recorded during the growing season. In 2019, 1168 measurements were removed, corresponding to 37.1% of the growing season period. The correction of fDOM signal to the temperature was performed at reference temperature (20°C), as proposed by de Oliveira et al. (2018).

During the 2018 and 2019 growing seasons, punctual water samplings were taken in the stream. At each sampling station, water samples were analysed for the DOC concentration and fDOM measurements taken with the EXO2 multi-parameter probe along the stream. The fDOM measurements were used to determine DOC, considering the relationship $f$(fDOM) = [DOC], where fDOM is the corrected signal fluorescence of DOM measured in quinine sulfate units (QSU) and [DOC] is the dissolved organic carbon concentration in mg C L$^{-1}$.

The first EXO2 multi-parameter probe that had been installed in June 2018 (calibration model I) was replaced with a new EXO2 multi-parameter probe in August 2018 which was used to the end of the data monitoring in May 2020 (calibration model II; Table SI.1). Each EXO2 multi-parameter probe was calibrated independently. Due to fouling (development of a biofilm on the surface of the sensor) of the fDOM sensor leading to a deviation of the calibration model, the calibration model was adjusted during the 2019 growing season and two more calibration models were developed to correct the fDOM deviation. The models are presented in Table SI.1.

#### 3.2.2 Stream hydrology

At the outlet of the stream, a 'V-shaped' weir was installed perpendicularly to the stream. The discharge was derived from the water level in the stream measured by an ultrasonic distance sensor (SR50, Campbell, USA) during the 2018 growing season and a water-level logger (U201-04, Hobo, Onset, USA) from June 2019, to replace the ultrasonic distance sensor, damaged during the spring freshet. The calculation method was described by Taillardat et al. (2022). The relation between the water level and the stream flow was calibrated using stream flow punctual measurements at the location of the water-level logger.



Due to uncertainty of measurements of stream discharge during the spring thaws, daily water discharge was modelled using the Peatland Hydrologic Impact Model (PHIM) developed by Guertin et al. (1987) and detailed by Riahi (2021).

### 3.2.3 Peatland hydrology

WTD was recorded hourly at the six wells (Fig. 1) equipped with a water-level data logger (HOBO, Onset, USA) for continuous measurement of WTD and temperature, from June 2018 to October 2020 as described in Prijac et al. (2022). In 2018, the water level loggers were U20-001-04 models (Hobo, Onset, USA) and replaced in 2019 with U20l-04 models (Hobo, Onset, USA). Those are slightly less precise (± 0.2% against ± 0.1% for the 2018 sensors) but better adapted to the meteorological conditions of the study site because of the battery durability for periods when temperatures are below 0°C. The sensors were placed into 200 wells, suspended with a measured metal wire and kept submerged (i.e. about -0.6 m below the peat surface). Another sensor was installed next to a rain gauge to record atmospheric pressure variability and to correct piezometer pressure.

Peat temperature was recorded hourly at 5, 10, 20 and 40 cm depths, using Hobo TMC50 temperature probes, coupled with Hobo U12-008 data loggers (Onset) from June 2018 to October 2020 as described in Prijac et al. (2022).

### 3.2.4 Rainfall

Rainfall was measured using a tilting bucket rain gauge (Onset, 0.2 mm). The bucket was connected to a sealed reed switch that produced a contact closure for every 0.2 mm of rainfall. Hourly measurements of rainfall consisted of the number of contacts resulting from 0.2 mm of rainfall.

### 3.3 Calculation of DOC exports

### 3.3.1 DOC concentration gap filling

Considering the percentage of removed fDOM signals (31.7% of the total measurements), a gap-filling method was performed on hourly DOC concentrations. The gap filling was conducted with a random forest model using a training data set containing the stream discharge record, water temperature, pH, dissolved oxygen saturation and specific conductivity (54.6% of the time series). The prediction of the data used by the random forest method (from the 'randomForest' package in R) was based on an unsupervised and nonparametric method of modelling. Models based on the validation dataset (13.7% of the time series) 215 presented a good fit between the observed and predicted DOC concentrations, with a correlation of 0.99 (p < 0.001); the mean squared residuals was 0.28 and the percentage of variance explained by the model was 98.7 % (Fig. SI.1). Modelled concentrations were included in the calculation of DOC exports. The importance of variables included in the random forest model is presented in Table SI.2. They were obtained using the argument 'importance' of the RandomForest function in R.

Gap filling of the DOC concentration could not be performed during the rest of the time series (i.e. non-growing 220 season) due to the bad quality of the model (i.e. low linear relationships between the predicted and measured values). In this case, the 10th quantile of the DOC concentration was used to fill the gaps.


### 3.3.2    Calculation of stream DOC exports

The DOC load at the outlet of the catchment (g C m$^{-2}$ year$^{-1}$) was calculated as in equation (1).

$$DOC_f = \frac{\sum_{i=1}^{n}[DOC]_i \times Q_i \times dt}{S} \tag{1}$$

In the above equation, $[DOC]_i$ corresponds to the DOC concentration in g m$^{-2}$ at step measurement $i$, $Q_i$ corresponds to the stream discharge in m$^3$ h$^{-1}$ at step measurement $I$, the variable d$t$ corresponds to the time step between high-frequency measurements and $S$ corresponds to the surface drained by the stream.

### 3.4  Analyses of flood event

### 3.4.1 Classification of time series in high- and low-flow periods to determine flood events

During the growing season, the hidden Markov model (HMM) was used to classify the time series into two states corresponding to the high- and low-flow periods (Kehagias, 2004; Guilpart et al., 2021) using the $R$ packages 'depmixS4' (Visser and Speekenbrink, 2010) and 'HiddenMarkov' (Harte, 2021). The distribution of probability to go from one state to another was

calibrated manually. After the HMM classification, the high-flow periods were manually adjusted to get a better integration of their beginnings. They were determined as the inflection of $Q$ before a persistent increase in this variable within a 12-h interval of a high-flow period determined by the HMM.

In addition, 12 individual flood events were manually isolated, six in 2018 and six in 2019 (Table SI.3) among the time series including DOC measurements of a satisfying quality (e.g. gap-filled DOC export values from the random forest

were excluded). Flood events were a subset of the total time series for individual analyses. They were identified by a two-letter code, the first letter corresponding to the year of the flood event (*A* for 2018 and *B* for 2019) and the second to the rank of the flood events each year, from *a* and following the alphabetical order.

### 3.4.2 Flood events characteristics

For each flood event, several descriptive and quantitative indicators were calculated; they are described in Table 1. During the

event, rainfall was summed up under the variable PP event. Rainfall was also summed up 2 days before the beginning of the event (AP2) and 14 days before the beginning of the event (AP14). The PP event and AP14 were added to obtain the variable PP total.

**Table 1.** List of variables used and their acronyms and units.

| Acronym | Variable | Units |
|---------|----------|-------|





| | | |
|---|---|---|
| AP14 | Antecedent precipitation 14 days before the beginning of an event | mm |
| AP2 | Antecedent precipitation 2 days before the beginning of an event | mm |
| $\beta$ | Index corresponding to the slope of the log-log DOC-Q relation during flood events (Godsey et al., 2009, 2019) | |
| DO mgL | Concentration of dissolved oxygen | mg L$^{-1}$ |
| DO sat | Saturation of dissolved oxygen | % saturation |
| DOC | Dissolved organic carbon | mg L$^{-1}$ |
| DOC lag time | Duration between the $Q$ peak and the DOC peak during a flood event | h |
| DOC$_{90}$ | Duration when 90% of maximum DOC concentrations were exceeded during a flood event | h |
| DOC$_{load}$ | Cumulative quantities of DOC exported to the stream per square metre during a defined time period | kg C m$^{-2}$ *time unit* |
| $\Delta$DOC | Difference between the initial DOC concentration at the beginning of the event and the peak DOC concentration | mg L$^{-1}$ |
| $\Delta Q$ | Difference between the initial discharge at the beginning of the event and the peak discharge | m$^3$ s$^{-1}$ |
| $\Delta$WTD | Difference between the initial WTD at the beginning of the event and the peak WTD | mm |
| FI | Flushing index, which corresponds to the difference between the DOC concentration at the peak of discharge and DOC concentration at the beginning of the event (Vaughan et al., 2017) | |
| HI | Hysteresis index, which corresponds to the difference between the normalized DOC concentration during the falling limb to an event and the rising limb to an event at an interval of 0.05 normalized $Q$ (Lloyd et al., 2016) | |
| PP event | Cumulative precipitation during a storm event | mm |
| P–Q lag time | Duration between the beginning of a precipitation event and the $Q$ increase at the beginning of a flood event | h |
| SPC | Specific conductivity | µS cm$^{-1}$ |
| $Q$ | Stream discharge | m$^3$ s$^{-1}$ |
| $Q$ lag time | Time elapsed between the beginning of the $Q$ increase and its peak | h |
| Total PP | Total catchment wetness corresponding to the sum of AP14 and the PP event | mm |
| WTD | Water table depth | m |

The P–Q lag time (in minutes) corresponds to the duration between the start of the rainfall and the $Q$ increase at the beginning of the event. The $Q$ lag time corresponds to the duration between the beginning of the event and the reaching of peak of $Q$ ($Q_{max}$). The DOC lag time corresponds to the duration between $Q_{max}$ and the peak of DOC (DOC$_{max}$). The DOC$_{90}$





corresponds to the period when 90% of $DOC_{max}$ was exceeded and can be summarized as the duration of the DOC plateau before the DOC concentrations decreased. The DOC load ($DOC_{load}$) was calculated as the DOC exports shown in equation (1) and corresponds to the quantity of DOC exported during the flood event. $DOC_{load}$ was divided by the event duration (in h) to provide a better comparison between events ($DOC_{load\ kgh}$).

The hysteresis index (HI), the flushing index (FI) and the $\beta$ index were determined from the relation between $Q$ and the DOC concentration. The HI was used to identify the hysteretic relation between DOC and $Q$ and corresponds to the difference in the integrals during the rising limb (i.e. the increasing phase of $Q$ during a high-flow event) and the falling limb (i.e. the decreasing phase of $Q$ during a high-flow event) of a high-flow event (Lloyd et al., 2016). HI values range from -1 for strong anticlockwise hysteretic relations to 1 for strong clockwise hysteretic relations; 0 indicates the absence of a hysteretic relation. The FI was calculated to describe the response of the DOC concentration during the rising limb of the flood (Vaughan et al., 2017). The FI ranges from -1 to 1; a value < 0 indicates that DOC is diluted during the rising limb while a value > 0 indicates accretion of DOC during the rising limb. The $\beta$ index corresponds to the slope of the logarithmic relation between $Q$ and the DOC concentration and provide information regarding the limiting factor of the DOC exports (Godsey et al., 2009). A $\beta$ index value < 0 indicates a source limitation of the DOC exports, a $\beta$ index value > 0 reveals that the DOC exports are transport-limited and $\beta$ = 0 indicates the DOC exports are chemostatic (Godsey et al., 2009, 2019; Zarnetske et al., 2018).

### 3.5 Statistical analyses

The data analyses were performed in R (CRAN-Project) and RStudio interface (RStudio Inc., USA). The figures were produced using the package 'ggplot2' (Wickham, 2016). Correlations between DOC and explanatory variables (porewater, air and stream temperature, $Q$, conductivity, pH, saturation of dissolved oxygen and dissolved oxygen concentrations) were evaluated using a multiple linear regression model. The p-values and Spearman correlation factors of individual variable effects on DOC concentrations were used as an indicator of model quality.

Prior to clustering the flood events, correlation and collinearity between variables were evaluated by measuring the variance inflation factor (VIF) function using the R package 'car'. Variables were removed when the correlation with another variable exceeded 0.8 and the VIF exceeded 5. The variables retained to perform clustering were the event duration, the minimum temperature, the average $Q$, the minimum WTD, the $\Delta DOC$, the HI index, the $\beta$ index, the FI, the initial WTD, the $Q_{max}$ and the $DOC_{load}$. Hierarchical clustering was performed based on principal component analysis (PCA) to classify each individual event into clusters. The number of clusters was determined according to the 'elbow method' as the optimal number of clusters corresponds to values when the inertia (i.e. the information given by additional clusters) decreases. The R package 'FactoMineR' was used for the PCA and hierarchical clustering.

The low- and high-flow periods were determined by using the HMM with the R package 'HiddenMarkov', which is designed for time series data. The HMM on log-transformed $Q$ (log$Q$) was performed based on hourly data.





## 4.    Results

### 4.1    High-frequency monitoring of hydrological variables and temperature


The maximum daily rainfall was 41 mm day$^{-1}$ in September 2018 (for the 2018–2019 period) and 39 mm day$^{-1}$ in August 2019 (for the 2019–2020 period). During the summer of 2018, the wettest month was July with total rainfall of 98 mm, while the wettest month during the summer of 2019 was August with 129 mm. The average WTD was -0.26 m and ranged from -0.09 to -0.43 m. The lowest WTD was in July and August 2019 with a monthly average of -0.30 ± 0.06 and -0.30 ± 0.07 m,

respectively. The average annual $Q$ was 0.0204 m$^3$ s$^{-1}$ in 2018–2019 and 0.0169 m$^3$ s$^{-1}$ in 2019–2020. During the growing season, the lowest monthly average discharge occurred in July of each year, with 0.0104 m$^3$ s$^{-1}$ in 2018–2018 and 0.0074 m$^3$ s$^{-1}$ in 2019–2020. In 2018–2019, the highest discharge was 0.0683 m$^3$ s$^{-1}$ measured in June 2018 and in 2019–2020 it was 0.10002 m$^3$ s$^{-1}$ measured in September 2019.

There was a strong positive exponential relationship between WTD and $Q$ ($rho$ = 0.82, p < 0.0001; Fig. 2a). This

nonlinear relationship suggests a threshold of WTD on lateral discharge generation. When low, WTD variations do not influence $Q$, which remains low. An increase in WTD above a threshold observed between -0.33 and -0.19 m leads to lateral discharge generation and an increase in $Q$ (Fig. 2a).

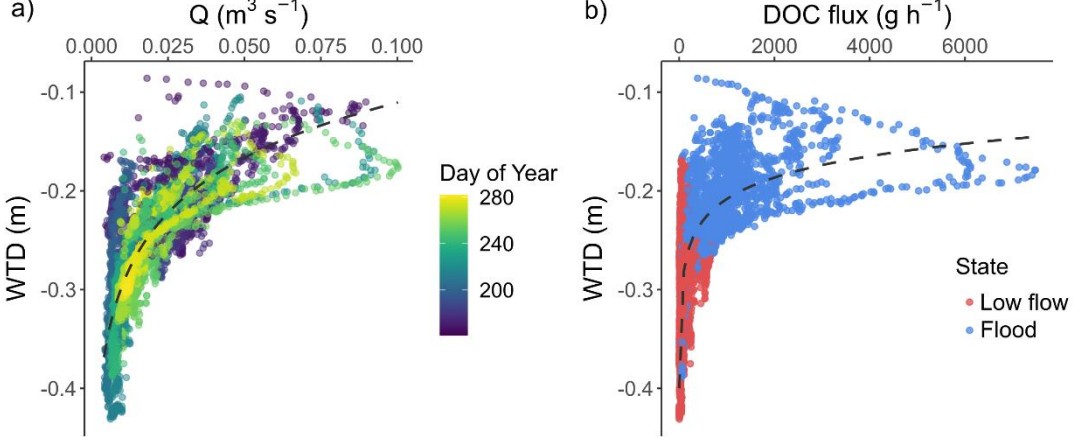

**Figure 2.** (a) Relation between hourly measurements of the water table depth (WTD, in m) and stream discharge ($Q$, in m$^3$ s$^{-1}$). The colour represents the day of the year and the dashed line corresponds to the logarithmic relation between WTD and $Q$. (b) Relation between the hourly measurements of WTD (m) and hourly DOC flux (g h$^{-1}$). The colour represents the hydrological state according to the hidden Markov model and the dashed line corresponds to the logarithmic relation between WTD and DOC flux.

The average peat porewater temperature was 11.5 ± 2.4°C and was very similar in 2018 (11.4 ± 2.6°C) and 2019

(11.7 ± 2.3°C). The warmest peat porewater temperature was 15.1°C measured in August 2019 and the coldest was 5.6°C





measured in June 2018. During the summer, the average monthly temperature in June increased from 7.1 ± 1.0°C in 2018 and 8.3 ± 0.8°C in 2019 to reach a maximum of slightly above 14°C in August. The temperature decreased in autumn, to a similar average October temperature (8.6 ± 0.4°C in 2018 and 8.7 ± 0.5°C 2019). The average water temperature recorded at the stream outlet was 13.2 ± 6.7°C. The average water temperature in 2018 was warmer, 13.9 ± 7.0°C compared with 12.7 ± 6.2°C

in 2019. As for the air temperature, the water temperature increased from about 11°C in June to 15.6°C and 16.9°C in July and August, respectively. The water temperature subsequently decreased in September, with similar values in both years (10.6 ± 3.5°C in 2018 and 10.2 ± 2.7°C in 2019).

## 4.2 DOC concentrations and exports from the peatland stream outlet

The average DOC concentration recorded at the peatland stream outlet was 6.3 ± 4.6 mg L$^{-1}$ and the median was 4.9 mg L$^{-1}$.

The maximum DOC concentration was 24.2 mg L$^{-1}$ in early August 2019 and the minimum was 0.9 mg L$^{-1}$ in September 2018 (Fig. 3g). Correlations between the DOC concentration and hydrological and physicochemical variables are presented in Table SI.2. The DOC concentration was significantly positively correlated with $Q$ and WTD (Table SI.2). DOC was positively correlated with water temperature but only when considering the complete period of measurements. The random forest model applied during the growing season data set highlighted the important contribution of hydrological variables (WTD and $Q$;

Table SI.2). It also showed the importance of the porewater temperature on the DOC stream concentration prediction, which was negatively correlated with DOC concentrations. During the growing season, the log-transformed hourly DOC exports were significantly correlated with $Q$ (cor = 0.79, $p < 0.0001$) and with WTD (cor = 0.75, $p < 0.0001$; Fig. 2b).









**Figure 3.** Times series of (a) stream and porewater temperature and precipitations, (c) water table depth (WTD), d) log-transformed stream discharge (log*Q*), (d) the dissolved organic carbon (DOC) concentration in the stream and e) DOC exports, from June 2018 to May 2020. Colours in the (b)–(e) correspond to the periods of flood (in blue) and low flow (in red). Grey vertical bars represent individual storm events. Yellow diamonds represent DOC concentration analyses from punctual sampling at the stream outlet.

We calculated the specific DOC exports from the peatland by using an approach based on distinction between the DOC sources during high flow and low flow. The assumption supporting this approach is that the peatland is the main contributor to DOC exports during high flow – because other sources are considered negligible – while during low flow, the most conservative approach is to consider the whole catchment as the potential DOC source. The surface considered in the specific DOC export calculation [*S* in equation (1)] is the catchment surface (2 219 574 m²) during low flow and the peatland surface (1 702 353 m²) during high flow.


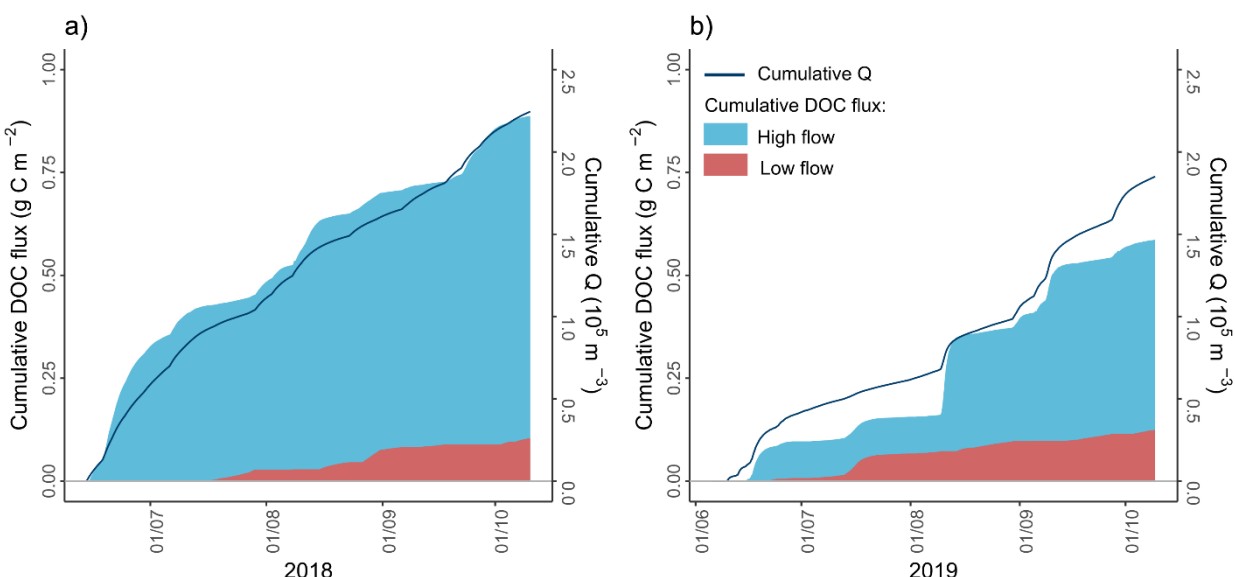

**Figure 4.** Cumulative dissolved organic carbon (DOC) flux (in g C m⁻²) and the cumulative stream discharge (in m³) during the (a) 2018 and (b) 2019 growing seasons. * The staircase trend observed in 2019 can be explained by long periods of drought with very low DOC concentration with discharge given the low DOC exports (Fig. 3e).

The specific annual DOC exports were 1.87 g C m⁻² y⁻¹ for June 2018–May 2019 and 1.27 g C m⁻² y⁻¹ for June 2019–May 2020 (Table 2 and Fig. 4). The strategy used to calculate the specific DOC exports by distinguishing high flow and low flow provides a better estimation of exports. If the most conservative surface (i.e. the catchment area) would have been used





to calculate the specific exports, it would have been 1.46 g C m$^{-2}$ y$^{-1}$ in 2018–2019 and 0.99 g C m$^{-2}$ y$^{-1}$ in 2019–2020. This approach provides a range for the plausible specific DOC exports from the peatland between 1.46 and 1.91 g C m$^{-2}$ y$^{-1}$ for 2018–2019 and between 0.99 and 1.29 g C m$^{-2}$ y$^{-1}$ for 2019–2020. During the period corresponding to the threshold of the 85th percentile of the $Q$ measurements (i.e. 15% of the total time series with the highest measured $Q$), the DOC exports represented 63.6% of the total exports during the 2018–2019 period and 66% during the 2019–2020 period.


**Table 2.** (a) Monthly specific dissolved organic carbon (DOC) flux (g C m$^{-2}$ month$^{-1}$) at the outlet stream from June 2018 to May 2020 and distinguishing flux during high flow when (1) the surface of the peatland is considered in the calculation and (2) the watershed is considered in the flux calculation. (b) Summary of DOC flux during the two growing seasons, the total recorded and their proportion during high- and low-flow periods.

(a)

|  | 2018–2019 DOC flux (g m$^{-2}$ month$^{-1}$) | | 2019–2020 DOC flux (g m$^{-2}$ month$^{-1}$) | |
| --- | --- | --- | --- | --- |
| Month | High flow | Low flow | High flow | Low flow |
| June | 0.452 | 0.000 | 0.102 | 0.008 |
| July | 0.130 | 0.022 | 0.000 | 0.009 |
| August | 0.167 | 0.053 | 0.229 | 0.016 |
| September | 0.144 | 0.011 | 0.327 | 0.012 |
| October | 0.208 | 0.003 | 0.080 | 0.005 |
| November | 0.208 | 0.003 | 0.099 | 0.000 |
| December | 0.000 | 0.010 | 0.060 | 0.001 |
| January | 0.000 | 0.003 | 0.000 | 0.010 |
| February | 0.000 | 0.004 | 0.000 | 0.008 |
| March | 0.000 | 0.006 | 0.000 | 0.010 |
| April | 0.052 | 0.008 | 0.136 | 0.001 |
| May | 0.418 | 0.000 | 0.157 | 0.000 |
| Total per conditions | 1.727 | 0.138 | 1.189 | 0.079 |
| Specific flux | 1.865 | | 1.268 | |

(b)

|  | 2018–2019 | | | 2019–2020 | | |
| --- | --- | --- | --- | --- | --- | --- |
|  | Proportion of measurements (%) | Flux (g m-2 y-1) | Proportion of flux (%) | Proportion of measurements (%) | Flux (g m-2 y-1) | Proportion of flux (%) |
| High flow | 59.1 | 1.727 | 92.6 | 44.1 | 1.189 | 93.8 |





| | | | | | | |
|---|---|---|---|---|---|---|
| Low flow | 40.9 | 0.138 | 7.4 | 55.9 | 0.079 | 6.2 |
| Total | 100.0 | 1.865 | 100.0 | 100.0 | 1.268 | 100.0 |

## 4.3 Analyses of flood events

### 4.3.1 Description of the flood events

Twelve flood events were isolated over the two growing seasons, six in 2018 and six in 2019 (see the grey vertical bars in Fig. 3). The average flood event duration was 4.8 ± 2.1 days. Aa was the longest event (10 days) while Ac was the shortest event (2 days; Table SI.3).

The Bd event had the lowest rainfall (8 mm) while the Bb event had the highest rainfall (34 mm). The antecedent rainfall 14 days before the beginning of the event was between 10 mm during the Ac event and 71 mm before the Be event. The maximum discharge during flood events varied from 0.026 m$^3$ s$^{-1}$ (Ac) to 0.1 m$^3$ s$^{-1}$ (Be). The discharge increase ($\Delta Q$) varied from 0.019 m$^3$ s$^{-1}$ (Ac) to 0.084 m$^3$ s$^{-1}$ (Bb). $\Delta$WTD during an event was between 0.08 m during the Ba event and 0.25 m during the Bb event. The DOC peak concentration varied from 5.0 mg L$^{-1}$ during the Ad event to 24.2 mg L$^{-1}$ during the Bb event. Regarding $\Delta Q$ and $\Delta$WTD, the Bb event also showed the highest DOC concentration increase ($\Delta$DOC, 22.5 mg L$^{-1}$). The Bb event also presented the highest hourly DOC exports (DOC$_{load}$), namely 3.14 kg h$^{-1}$. The Bf event had the lowest DOC$_{load}$ at 0.23 kg h$^{-1}$.

The HI was always negative, associated with anticlockwise hysteresis, except for the Ba event that had an HI of 0.05, indicating the absence of a hysteretic relation between $Q$ and DOC (Fig. 5). The HI varied from -0.16 for the Bf event to -0.56 for the Ae event. The $\beta$ index was always positive, indicating a constant transport limitation of DOC during flood events. The Af event showed a FI of 0.02, reflecting the absence of change in the DOC concentration between the beginning of the event and the peak of $Q$. The positive FI for the other events indicated that the DOC concentration increased during the rising limb of the hydrograph and was between 0.25 for the Ae event and 0.98 for the Bb event.

The shortest lag time between the rainfall and the beginning of the $Q$ increase (P–Q lag time) occurred during the Ba event (2 h). The longest P–Q lag time was during the Bc and Bd events (7 h). The $Q$ lag time ranged from 15 h for the Ac event to 39 h between the beginning of the event and $Q$ peak during the Bc event. The DOC lag time or the lag time between the peak of Q and the peak of DOC ranged from 7 h during the Ac event to 36 h during the Ad event. The shortest DOC$_{90}$ occurred during the Ac event (2 h), while the longest DOC$_{90}$ was 17 h during the Ae event.

### 4.3.2 Classification and typology of flood events

The hierarchical clustering performed on based PCA (presented in Fig. SI.2) classified the flood events into three groups (Fig. 6a). Cluster 1 included the Ab, Ac, Ad, Ae, Af, Bc and Bd events; cluster 2 comprised the Ba, Be and Bf events; and cluster 3 included the Aa and Bb events.





**Figure 5.** The hysteretic relations between the normalized stream discharge ($Q$) and the normalized dissolved organic carbon (DOC) for the events of (a) cluster 1, (b) cluster 2 and (c) cluster 3. The colour represents the count of the measure, from 0 at the beginning of the event to the end. The hysteresis index (HI), the flushing index (FI) and the $\beta$ index are presented for each event.


The average variable values by cluster are summarized in Table 3. The events of cluster 3 had greater DOC exports, namely $2.4 \pm 0.1$ kg C h$^{-1}$, compared with clusters 1 and 2 ($0.6 \pm 0.3$ and $1 \pm 2.1$ kg h$^{-1}$, respectively). The events of cluster 3 also had the highest DOC$_{max}$ and $\Delta$DOC of $19.4 \pm 2.1$ mg C L$^{-1}$ and $15 \pm 3.7$ mg C L$^{-1}$, respectively. By contrast, the events of





cluster 2 presented the lowest average $DOC_{max}$ ($8 \pm 13.7$ mg C $L^{-1}$), but the events of cluster 1 presented the lowest $\Delta DOC$ (6.4

$\pm 4.1$ mg C $L^{-1}$).

Although the events of cluster 1 had the highest $\Delta DOC$, the events of cluster 2 had the highest $Q_{max}$ and $\Delta Q$, namely $0.086 \pm 0.018$ and $0.065 \pm 0.022$ $m^3$ $s^{-1}$, respectively. $Q_{max}$ and $\Delta Q$ for the events of cluster 3 were $0.081 \pm 0.001$ $m^3$ $s^{-1}$ and

$0.062 \pm 0.010$ $m^3$ $s^{-1}$, respectively. The events of cluster 1 had the lowest $Q_{max}$ and $\Delta Q$ of $0.043 \pm 0.012$ and $0.029 \pm 0.011$ $m^3$ $s^{-1}$, respectively. The events of cluster 3 showed the lowest $WTD_{initial}$ ($-0.31 \pm 0.07$ m) and the highest $WTD_{max}$ ($-0.11 \pm 0.01$ m) and thus the highest $\Delta WTD$ ($0.19 \pm 0.08$ m). The events of cluster 2 presented the lowest $\Delta WTD$ ($0.09 \pm 0.11$ m) and the highest $WTD_{initial}$ ($-0.21 \pm 0.09$ m). Conversely, the events of cluster 1 showed a low $WTD_{initial}$ ($-0.30 \pm 0.06$ m) and despite a relatively high $\Delta WTD$ of $0.15 \pm 0.05$, they reached the lowest average $DOC_{max}$ ($-0.15 \pm 0.02$ m).




**Figure 6.** (a) Representation of the hierarchical clustering performed based on principal component analysis discriminated the events into three clusters (Cluster 1 = yellow, Cluster 2 = red, Cluster 3 = blue). (b) For each event, the variables have been mean centred and averaged by cluster. The representation of averaged mean-centred values allowed us to identify the behaviour of variables in each cluster.



On average, the events of cluster 1 presented the lowest HI ($-0.4 \pm 01$) while the events of cluster 2 showed the highest HI ($-0.1 \pm 0.1$). The events of clusters 1 and 2 shared a similar $\beta$ index of 0.5, while the events of cluster 3 had the highest $\beta$ index ($0.8 \pm 0.1$). The events of cluster 3 had the highest FI ($0.8 \pm 0.1$), compared with $0.6 \pm 0.2$ for the events of cluster 2 and 385 $0.3 \pm 0.3$ for the events of cluster 1.

**Table 3.** Summary of the variables and indexes (presented as mean $\pm$ standard deviation) for each cluster of flood events. The variables include the duration of events; the average stream temperature (T°C); the initial, maximum and change in ($\Delta$) the stream discharge ($Q$); the water table depth (WTD); and the dissolved organic carbon (DOC) concentration. The hysteretic index (HI), flushing index (FI) and $\beta$ index characterize the storm events. Precipitation variables comprise the total precipitation during events (PP event) and antecedent precipitation 2 days (AP2) and 14 days (AP14) prior to the beginning of an event. Total PP corresponds to the sum of AP14 and PP events. The P–Q lag time corresponds to the duration between a precipitation event and the beginning of the increase in $Q$. The $Q$ lag time corresponds to the duration between the beginning of the discharge increase and the discharge peak. The DOC lag time corresponds to the duration between the discharge peak and the DOC peak. $DOC_{90}$ corresponds to the period when 90% of the maximum DOC concentration was exceeded.

|  | **Cluster 1** | **Cluster 2** | **Cluster 3** |
|---|---|---|---|
| Duration (day) | $3.9 \pm 1.3$ | $5.2 \pm 1.5$ | $7.3 \pm 0.2$ |
| Stream T°C min (°C) | $6.6 \pm 3.6$ | $5.6 \pm 5.3$ | $7.9 \pm 1.3$ |
| Stream T°C max (°C) | $15.6 \pm 3.3$ | $15.2 \pm 5.2$ | $19.5 \pm 1$ |
| Stream T°C average (°C) | $11.1 \pm 3.1$ | $9.5 \pm 4.7$ | $12.9 \pm 0.6$ |
| Porewater T°C min (°C) | $11.3 \pm 1.9$ | $9.1 \pm 1.7$ | $10.3 \pm 5.7$ |
| Porewater T°C max (°C) | $12.1 \pm 1.8$ | $10.2 \pm 2.0$ | $11.4 \pm 4.9$ |
| Porewater T°C average (°C) | $11.8 \pm 1.9$ |  | $11.0 \pm 5.2$ |
| $Q$ initial (m³ h⁻¹) | $55.9 \pm 23.1$ | $74.3 \pm 12.5$ | $66.3 \pm 39.4$ |
| $Q_{max}$ (m³ h⁻¹) | $153.8 \pm 41.9$ | $308.1 \pm 65.9$ | $289.8 \pm 3.7$ |
| $\Delta Q$ (m³ h⁻¹) | $98 \pm 44.4$ | $233.9 \pm 78.5$ | $223.5 \pm 35.7$ |
| cumulative $Q$ (m³ h⁻¹) | $9562 \pm 3036$ | $19145 \pm 790$ | $29184 \pm 835$ |
| $WTD_{initial}$ (m) | $-0.30 \pm 0.06$ | $-0.21 \pm 0.09$ | $-0.31 \pm 0.07$ |
| $WTD_{max}$ (m) | $-0.15 \pm 0.02$ | $-0.12 \pm 0.02$ | $-0.11 \pm 0.01$ |
| $\Delta WTD$ (m) | $0.15 \pm 0.05$ | $0.09 \pm 0.11$ | $0.19 \pm 0.08$ |
| DOC initial (mg L⁻¹) | $3.5 \pm 1.8$ | $5.6 \pm 2.9$ | $3.7 \pm 0.6$ |
| DOC max (mg L⁻¹) | $10.3 \pm 4.2$ | $12.8 \pm 8.8$ | $18.7 \pm 3.1$ |
| $\Delta DOC$ (mg L⁻¹) | $6.8 \pm 3.8$ | $7.3 \pm 11.8$ | $15 \pm 3.7$ |





| | | | |
|---|---|---|---|
| HI | -0.4 ± 0.1 | -0.1 ± 0.1 | -0.3 ± 0.1 |
| $\beta$ | 0.6 ± 0.2 | 0.5 ± 0.1 | 0.8 ± 0.1 |
| FI | 0.4 ± 0.3 | 0.6 ± 0.2 | 0.8 ± 0.1 |
| PP event | 19 ± 9 | 16 ± 12 | 34 ± NA * |
| AP2 (mm) | 6 ± 6 | 12 ± 1 | 20 ± NA * |
| AP14 (mm) | 34 ± 19 | 42 ± 11 | 42 ± NA * |
| TotalPP (mm) | 53 ± 15 | 58 ± 24 | 76 ± NA * |
| P–Q lag time (h) | 4.7 ± 2 | 3.3 ± 0 | 5 ± NA * |
| Q lag time (h) | 23.7 ± 8.2 | 26 ± 1.4 | 28.5 ± 14.1 |
| DOC lag time (h) | 24.1 ± 12.3 | 10.7 ± 3.5 | 11 ± 14.8 |
| $DOC_{90}$ (h) | 9.7 ± 4.9 | 7 ± 4.2 | 11.5 ± 2.1 |
| DOC load (kg) | 71.1 ± 36.4 | 161.4 ± 145.5 | 370.1 ± 23.2 |
| DOC load (kg h$^{-1}$) | 0.8 ± 0.4 | 1.6 ± 1.3 | 2.1 ± 0.3 |

*As no precipitation data was available for the Aa event, it was not possible to calculate a standard deviation for the events of cluster 3. The values correspond of the results for the Bb event.

For cluster 3, the rainfall data were only available for the Bb event. However, this event showed the highest total rainfall (76 mm), supported by the highest rainfall during the events and high rainfall before the event. The lowest rainfall before the events occurred for cluster 1 and the rainfall during the events of 19 mm on average led to the lowest total PP of 53 ± 15 mm, which was slightly lower than events of cluster 2 (58 ± 24 mm).

## 5.    Discussion

### 5.1 Peatland hydrological connectivity controls DOC exports to the stream

Coupling high-frequency monitoring of DOC concentrations with hydrological measurements ($Q$ and WTD) was important to better understand the relationships between DOC concentration dynamics at the outlet and the hydrological functioning of the peatland. In the studied peatland, we observed a control of hydrological variables (i.e. WTD and $Q$) on the DOC concentrations at the stream outlet (Table SI.2). The increase in WTD led to an increase in $Q$ and DOC concentrations at the outlet and,
consequently, to an increase in DOC exports (Fig. 2). DOC mobilization during high-flow periods exhibited anticlockwise hysteresis (Fig. 5), reflecting the pronounced connectivity between DOC-rich sources within the catchment and the stream (Tunaley et al., 2017). The positive FI and $\beta$ index (Table 3 and Fig. 5) indicate accretion of DOC during flood episodes and reveal a transport limitation of DOC (Vaughan et al., 2017; Zarnetske et al., 2018).

The logarithmic relationship between WTD and $Q$ (Fig. 2a) highlights the crucial contribution of peatland during
high-flow periods. This mechanism has been described as the threshold of runoff and subsurface flow generation effect induced by a greater WTD (Frei et al., 2010) based on the transmissivity feedback mechanism (Bishop et al., 2004) and leading to $Q$



increase. It also illustrates the coupling of WTD and DOC exports (Fig. 2b), which are favoured by subsurface flows of water into DOC-rich horizons and less decomposed peat (Austnes et al., 2010) initiated by a rainfall event leading to the increase in WTD and confirmed by a significant positive correlation between DOC exports and WTD (cor = 0.75, p < 0.0001; Fig. 2b).

An increase in the subsurface flows has been described as the dominant hydrological control on DOC mobilization and exports to peatland streams (Bishop et al., 2004; Birkel et al., 2017; Rosset et al., 2022). In addition, the fluctuating water table in the acrotelm enhances the DOC available to the lateral discharge during high-flow events (Kalbitz et al., 2000; Worrall et al., 2002; Grand-Clement et al., 2014). During the driest periods, the DOC diffuses through the peat and becomes available for further mobilization through lateral discharge during rewetting of the acrotelm (Worrall et al., 2008). This is consistent with

the particularly important DOC exports measured during the summer of 2019 (the Bb event, Fig. 3), after July 2019, which was the driest month (27 mm of precipitation). As the increase in the DOC concentration and exports in the stream followed the WTD increase (Figs. 2 and 3), we assume that the DOC exported during high flow is mainly derived from leaching of the acrotelm.

The intermittence of DOC concentration peaks showed that DOC exports are constrained during flood episodes,

which are characterized by rapid and significant increases in WTD and $Q$ (Fig. 2). As DOC concentration variations and exports and hydrological variables are closely related, the shift from low- to high-flow periods can be viewed as the hydrological reconnection between peat – that is, the DOC reservoir – and the peatland drainage stream (Billett et al., 2006).

## 5.2 The succession of low and high flow determines specific peatland DOC exports

In contrast to the assumption that the peatland is the main source of exported DOC during high-flow periods, we found that

the hydrological connection between the peat and the stream is less clear during the low-flow periods (Fig. 3). Consequently, we used an alternative approach to calculate specific DOC exports by using two different catchment surface areas, depending on the discharge. Based on the classification of the discharge in high- and low-flow periods, we calculated the specific exports of the peatland as the amount of DOC exported during the high-flow periods. During the low-flow periods, we used the more conservative approach; specifically, we used the total catchment area as the surface reference for the export calculation during

those periods.

We argue that this pragmatic approach provides a more accurate estimation of the specific DOC exports from the peatland. The annual exports using this approach were 1.86 g C m$^{-2}$ y$^{-1}$ in 2018–2019 and 1.27 g C m$^{-2}$ y$^{-1}$ in 2019–2020. Approaches using the whole catchment area during the study period would have underestimated the exports by 21.6% in 2018–2019 and by 21.8% in 2019–2020. Conversely, using the peatland area within the catchment to calculate DOC exports would

have overestimated the exports by only 2.2% and 1.9% for 2018–2019 and 2019–2020, respectively, because the peatland covers 76.7% of the whole watershed. This approach could be easily implemented in future work using high-frequency measurements of DOC and $Q$, allowing to determine specific peatland DOC exports. While the high-flow periods accounted for 59% and 44% of the complete time series in 2018–2019 and 2019–2020, respectively, the specific exports accounted for 92.6 % of the annual exports in 2018–2019 and 93.8% in 2019–2020 (Table 2b). This approach supports the dominance of the





peatland contribution in annual DOC exports (Tipping et al., 2010) and the importance of high flow as key moments of DOC exports (Rosset et al., 2019), particularly by the increase of the hydraulic connectivity between the peatland and the stream (Birkel et al., 2017; Tunaley et al., 2017).

       Annual peatland carbon exports present high differences between years considering the exports for 2018–2019 were 1.5 times higher than for 2019–2020. Important annual variation of DOC exports was previously observed in peatland drainage
stream from a factor 1.6 to 3 and attributed to interannual variation of the discharge (Worrall et al., 2009; Dinsmore et al., 2013; Leach et al., 2016; Birkel et al., 2017; Rosset et al., 2019). The variability in the cumulative discharge at the stream outlet, 1.26 times higher in 2018–2019 compared with 2019–2020, explains the variations in DOC exports between the two years (Fig. 4).

       The exports measured in this study are lower than those previously measured in undisturbed boreal peatland drainage
streams, which varied from 3.7 to 18.0 g C m$^{-2}$ y$^{-1}$ (Köhler et al., 2008, 2009; Juutinen et al., 2013; Leach et al., 2016). The difference in DOC exports measured in the present study can be explained by different reasons. In our study site, the $\beta$ index clearly indicates a transport limitation of DOC during flood events (Fig. 5). The exports can be explained by the configuration of the catchment, and particularly by the sand deposits located on the west side of the peatland (Fig SI.2). Those deposits might limit the connectivity between the peat and the stream and reduce the lateral exports in this section. Second, most studies have
used discrete sampling to determine DOC exports (Köhler et al., 2008; Juutinen et al., 2013). Variability in DOC concentrations (Fig. 3) might lead to an overestimation of DOC based on discrete sampling.

       Taillardat et al. (2022) estimated the stream carbon GHG ($CO_2$ and $CH_4$) aquatic exports as 8.08 g C m$^{-2}$ y$^{-1}$ at our study site. DOC only accounts for 13.6%–18.8% of the total aquatic carbon exports. This is lower than previous studies that estimated the DOC contribution to aquatic carbon exports which have reported a contribution of greater than 50% (Roulet et
al., 2007; Worrall et al., 2008; Holden et al., 2012; Dinsmore et al., 2013; Leach et al., 2016). A hypothesis for the low contribution of DOC to aquatic carbon exports could be mineralization in the hyporheic zone (i.e. the soil-stream interface) where DOC can be intensively transformed into $CO_2$ (Krause et al., 2011; Lapierre and del Giorgio, 2014; Fasching et al., 2015; Rasilo et al., 2017; Taillardat et al., 2022). The low DOC exports measured at our site tend to moderate the importance of carbon loss by lateral exports.

**5.3 Variability in DOC lateral transfer patterns and implications in annual DOC exports**

The division of flood events between three clusters helped us understand the mechanisms leading to the different magnitudes of DOC exports (Table 3 and Fig. 6b). The events of cluster 1 seem to represent the most common type of flood events as it included 7 of the 12 events and accounted for 47.7% of the total event duration but with the lowest DOC$_{load}$ of 0.6 ± 0.3 kg C h$^{-1}$ (Table 3). They were marked by a low WTD$_{initial}$ (-0.30 m) and despite the slightly higher ΔWTD than the average (Fig.
6b), the lateral discharge did not lead to an important increase in $Q$ compared with the other clusters (Table 3). In addition to the low Δ$Q$ and $Q_{max}$, the low FI (Table 3) reflects low accretion of DOC (Vaughan et al., 2017). While Tunaley et al. (2017) interpreted that a low HI reflects a DOC source distant from the stream, in our study site, it seems more related to progressive





rewetting of the peat and slow lateral discharge leading to slow DOC mobilization to the stream (Bishop et al., 2004; Blaurock et al., 2021). Those conditions restricted the connectivity between DOC sources and the stream leading to low DOC loads (Fig.

7a).

a)

b)

c)

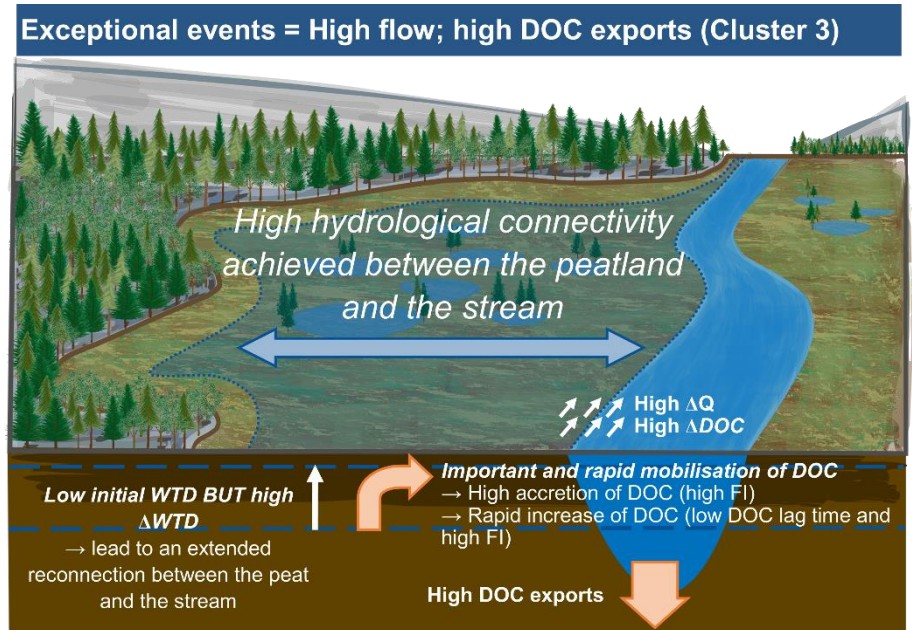

**Figure 7.** Theoretical models of flood events from (a) events of low flow and low dissolved organic carbon (DOC) loads (cluster 1), (b) events of high flow and average DOC loads (cluster 2) and (c) events of high flow and high DOC loads (cluster 3).

Cluster 2 comprised three events that occurred during the early and late growing season of 2019 (Fig. 6a). Those events had a comparable rainfall amount but a higher $\Delta Q$ compared with the events of cluster 1 (Table 3). The high $WTD_{initial}$ might indicate that these events succeeded a previously 'wet' period. This can be supported by the higher FI than events of the cluster 1 and the highest HI (Fig. 5). It reflects rapid DOC mobilization, simultaneously to the $Q$ increase, and from sources close to the stream (Tunaley et al., 2017; Blaurock et al., 2021). Those events might represent rapid flushing of DOC promoted by the high $WTD_{initial}$ and supported by the lowest $DOC_{90}$ leading to moderate DOC loads of 1.0 kg C h$^{-1}$ on average (Fig. 7b). Although the threshold of the lateral discharge generation was easily exceeded, the high HI suggests that DOC was mostly exported from sources close to the stream (Tunaley et al., 2017).

Cluster 3 comprised two events that occurred during early summer (Aa) and mid-summer (Bb) where the highest $\Delta WTD$ and $\Delta DOC$ and high $\Delta Q$ led to the highest $DOC_{max}$ (Fig. 6b). Consequently, during those events $DOC_{load}$ was 2.4–4 times higher than events of cluster 1 and cluster 2 respectively. Despite the low $WTD_{initial}$ of -0.31 cm comparable to the events of cluster 1, those events presented greater DOC exports. These findings indicate that $DOC_{load}$ is more constrained by the magnitude of the WTD increase rather than the initial WTD considering that WTD drawdown could stimulate the DOC production (Grand-Clement et al., 2014). During those events, the large WTD increase favoured the rapid circulation of water through the DOC-rich acrotelm (Inamdar et al., 2004) and supported by the high FI, indicating rapid flushing of DOC to the stream (Table 3). In addition, the anticlockwise hysteresis (HI of -0.3 in average, Table 3) highlights the extensive connectivity





between DOC sources within the peatland and the stream (Pellerin et al., 2012; Tunaley et al., 2016), supporting the high DOC exports (Fig. 7c).

Cluster 3 events appear to be extreme and associated with events with a low probability of occurrence. DOC exported during those events contributed to 24.3% and 24.4% of the total exports while only representing 8.5% and 3.8% of the growing season 2018 and 2019, respectively (Table 2b). These data suggest that the magnitude of an single event is at least as important as several events (Raymond and Saiers, 2010). Interestingly, those events did not happen during the same periods, revealing different export mechanisms. The Aa event occurred at the end of the spring freshet, which is known as an important period

of DOC exports (Tiwari et al., 2018). Conversely, the Bb event occurred during the warmest registered period, in August 2019, after 42 days of low flow and without a significant period of DOC exports between 26 June and 8 August 2019 (Figs. 3 and 4). A large amount of DOC was exported during high-flow events occurring throughout the warm periods. This may coincide with conditions that have previously been described as favourable for DOC production within the peat (Clark et al., 2007, 2009; Dinsmore et al., 2013). Then, the large rainfall events occurring before the event initiate an important WTD increase

that leads to DOC mobilization (Table SI.3; Grand-Clement et al., 2014; Zhu et al., 2022).

## 6.  Conclusion

Our study, measuring continuous DOC exports from a boreal peatland in north-eastern Canada, provides the very first insight concerning peatland DOC exports for this region. The use of high-frequency monitoring of hydrological variables and DOC concentrations has provided a comprehensive understanding of the temporal dynamics of DOC exports and the mobilization

patterns of DOC in a boreal peatland. The relationship between WTD and $Q$ highlights the major contribution of peat subsurface flows to $Q$ during flood events. Our data suggest that during these events, the exported DOC is mainly leached from the peatland. While the determination of specific DOC exports from the peatland remains a challenge, here we have proposed a time series analysis split between low- and high-flow periods. During the flood periods, the surface considered in the export calculations is the peatland area within the catchment. By contrast, during the low-flow periods, the catchment area

is considered the conservative surface reference in the calculation given the lack of a direct link between peat porewater discharge and DOC exports from the stream. DOC exported during high flow represented 92.6% and 93.8% of the total DOC exports during 59% and 44% for the 2018–2019 and the 2019–2020 periods, respectively. In addition, the use of a simple catchment surface in the export calculation would underestimate the exports by 22% compared with the new approach we have proposed.

The study of DOC mobilization during flood events supports the idea that variations in WTD generate lateral discharge that controls the magnitude of DOC being exported from the stream. Based on hierarchical clustering, three types of events were characterized with contrasting wetness conditions. The most common events (cluster 1) had a low $WTD_{initial}$ and a small WTD increase that limited the extent of the connectivity between the DOC sources and the stream. Conversely, the events of cluster 3 showed an important WTD increase, easily exceeding the threshold of runoff generation to facilitate DOC





mobilization and to increase its transfer through the stream. Those exceptional events can represent up to 24% of the total DOC exported during periods, accounting for 8% and 3% of the growing season in 2018 and 2019 respectively. The cluster 2 events represented intermediate conditions. While during those events the threshold of runoff generation was easily exceeded, previous events might have depleted DOC available to be transferred to the stream. This event presented relatively low DOC loads despite the high peak WTD and $Q$.

The response of DOC mobilization to hydroclimatic conditions in peatland is a key element in the magnitude of DOC exports. With climate change, precipitation regimes, the ice-free season duration and the water balance of ecosystems will be affected. Consequently, an increase in drought followed by intensive rainfall in the context of climate change might increase the aquatic DOC exports in boreal peatlands.

**Data availability**

The data have been submitted to a reliable repository and a DOI will be included in the manuscript.

**Author contributions**

Conceptualisation: Laure Gandois, Michelle Garneau, Antonin Prijac and Pierre Taillardat

Data curation: Antonin Prijac and Pierre Taillardat

Data analyses: Alex Ponçot, Antonin Prijac, Khawla Riahi and Pierre Taillardat

Formal analyses: Antonin Prijac and Laure Gandois

Funding acquisition: Michelle Garneau and Alain Tremblay

Investigation: Laure Gandois, Antonin Prijac and Pierre Taillardat

Methodology: Marc-André Bourgault, Laure Gandois and Antonin Prijac

Data collection: Antonin Prijac, Khawla Riahi and Pierre Taillardat

Writing – original draft preparation: Antonin Prijac

Writing – review and editing: Marc-André Bourgault, Laure Gandois, Michelle Garneau, Antonin Prijac, Khawla Riahi, Pierre Taillardat and Alain Tremblay

**Competing interest**

The authors declare that they have no conflict of interests.





**Acknowledgments**

The funding for this research was provided by the Natural Sciences and Engineering Research Council of Canada and Hydro-Quebec to Michelle Garneau (RDCPJ 514218-17). We thank Frederic Julien, Virginie Payre-Suc and Didier Lambrigot, from Laboratoire Ecologie Fonctionnelle et Environnement (UMR 5245 CNRS – UT3 – INPT, France), for performing DOC/TN and cation/anion analyses. Thanks to Dr Roman Teisserenc (ENSAT, Toulouse, France) and students Charles Bonneau, 555 Charles-Élie Dubé-Poirier, Camille Girard, Pénélope Germain-Chartrand, Léonie Perrier, Guillaume Primeau, Khawla Riahi and Karelle Trottier for their assistance in the field.

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
