# Peer review of "Hydrological connectivity controls dissolved organic carbon exports in a peatland-dominated boreal catchment stream"

_Hydrology and Earth System Sciences, 2022_

## Author Comment (AC1)

**Response letter to the reviewers: Manuscript hess-2022-426**

In this response letter, the reviewer's comments are in ***italic bold black***, our responses are in blue and significant new text added to the manuscript are in *italic green*. Changes made in the manuscript are tracked and line number referred to the revised manuscript.

**Reviewer #1**

***General comments***

***The authors collected a valuable data set on DOC export in small catchment, which is partly covered by peat. They highlighted the relationship between water table depths and DOC export, calculated area specific DOC export and analyzed DOC export mechanisms during individual flood events. Their data contributes to the ongoing discussion on mechanisms for DOC mobilization and its link to hydrological connectivity. I generally think that the paper is worthy for publication. However, there are a few points that need to be carefully addressed, especially regarding the calculation of area specific DOC export and peatland coverage.***

We thank the reviewer for the constructive comments that will help to improve the manuscript. Also, we are pleased the reviewer recognize the overall quality of the study presented here. We will make sure to carefully respond to the comments you addressed.

***A large part of your paper focuses and the difference between peat-covered and not peat-covered area. In L128 you mention the surface, however, you do not explain how you assessed peat coverage in your catchment. I think this would be very important regarding how important these areas are for your argumentation.***

You mentioned an important omission in our manuscript which was added in the study site. The peatland surface area was determined by GIS software and based on a lidar data.

*Line 130-138: "The catchment and peatland areas were determined through ArcGIS Pro 2.8.0 based on LiDAR images taken in 2004 (source: Hydro Quebec) and an aerial image from "World Imagery ArcGIS" taken on 8 May 2017 (resolution of 0.5m). The LiDAR images and generated databases were used by extrapolation to determine the Digital Elevation Model (DEM). The tools "flow accumulation" and "watersheds" in ArcGIS Pro 2.8.0 were used to generate the hydrological network and associated catchment area. A supervised classification of vegetation was conducted to delineate the peatland ecosystem boundaries within the catchment using the tools "create signatures" and "maximum likelihood classification". Lidar data covering the study site and provided by Hydro-Québec.*

***Also, your argumentation in section 5.2. is unclear to me. Of course, area specific DOC export increases with decreasing surface area. To switch between different surface areas according to flow, you need to be very sure of which area actually contributes to DOC export. In L424-430 you argue that peatlands are probably contributing less to DOC export during low flow periods because of a missing hydrological connectivity, so you use the total surface area to calculate area specific DOC export. You argue that during high flow the peatlands become more important for DOC export, therefore you use the smaller peatland-covered surface to calculate specific DOC export. But in my point of view, this leads to an overestimation of specific DOC***

*export. How can you be sure that the rest of the area does not contribute to DOC during high flow? Would a higher hydrological connectivity not lead to a larger contributing area rather than a smaller one? This needs to be made clear. In this context, it would be useful to understand where the peats are located. Are they further away from the stream (this would be unusual) and therefore connected only during high-flow? Would it be possible to highlight peat-covered areas in Figure 1?*

In your comment, you discussed about the pertinence of the specific DOC export calculation (by switching between peatland surface area and catchment area depending on hydrological conditions of high flow and low flow respectively).

First, the hypothesis is not that "peatland contributing less to DOC exports during low flow" but, because the hydrological connectivity between the peat and the stream is not important as during high flow (see Fig. 2 and in section 5.1, line 399-402 of the submitted manuscript) we chose the most conservative surface in the specific DOC export calculation (i.e., catchment surface area). We clarified this point in the revised manuscript.

*Line 472-474: "This approach relates to the hypothesis that DOC exported during high-flow is mainly derived from the peat lateral export while during low-flow, the hydrological connectivity between the peat and the stream is not clear (Fig. 2.a)."*

Second, you mentioned that as the hydrological connectivity is supposed to be larger during high-flow periods, the contributing surface should be larger. In our situation, the challenge is to determine the specific contribution of the peatland (within the catchment). We agree that the lack of source investigation cannot exclude the consideration of other sources than peatland during high-flow periods. However, studies on *C-Q* relationships conducted into mixed catchments, or catchment with a small surface covered by wetlands (including peatlands), showed that patterns of DOC mobilization observed in our study site brings specific features of mobilization patterns attribute to wetlands. Then, 5.2 section of the manuscript was enhanced to clarify our argumentation.

*Line 474-494: "Although the absence of DOC sources investigation within the catchment, the C-Q relationships might help understand DOC sources through hypothesis made on peatland lateral flow pathways within the catchment. During the studied floods episodes, C-Q relationships exhibited homogeneous pattern characterized by anticlockwise hysteresis and increases in DOC concentrations during the rising limb of the flood (Fig. 5). We previously interpreted them as the subsurface runoff in the DOC-rich acrotelm, caused by the rise of the water table (see section 5.1) and leading to the progressive reconnection between peat-derived DOC sources and the stream during flood events (Tunaley et al., 2016).*

*Understanding the DOC lateral transfer pathways is important to resolve the challenge of characterizing DOC sources and to estimate the contribution of forested soils which covered 17% of the studied site. In a mixed headwater catchment covered by only 22% of peatlands in riparian zones, Dick et al. (2015) estimated that 84% of exported DOC was derived from peat soils. In catchments dominated by mineral forested soils, Raymond and Saiers (2010) observed clockwise hysteretic loops, caused by the progressive depletion of available soil-derived DOC during the rising limb of the flood. Contrastingly, anticlockwise hysteretic loop combined with an increase of DOC concentrations during the rising limb was also observed from forested catchment. Despite the*

*dominance of forested area, authors attribute those relations to the contribution of riparian wetlands to DOC exports (Pellerin et al., 2012; Strohmeier et al., 2013). In our site, forested areas are mainly located on the west border of the catchment and some patches are in upstream sections*

85   *of the stream, while riparian areas in the downstream section is composed of peat (Fig. 1). This tends to even more moderate the importance of forested area in DOC exports contribution."*

*We argue that this pragmatic approach provides a more accurate estimation of the specific DOC exports from the peatland, although it generates a small overestimation since DOC export from other land covers are assumed to be negligible."*

90   Finally, as you suggested, the figure 1 was changed by a map distinguishes the surface area of the catchment covered by peat i.e., peat vegetation (and other components of the catchment such as forested areas i.e., non-peat vegetation, sand deposits and pools). This representation should provide a better view of the peatland position within the catchment, following your suggestion.

[Figure]

**Figure 1.** Map of the land cover of the Bouleau catchment which distinguishing areas covered by the drainage stream, sand deposits, pools, non-peat vegetation and peat vegetation.

95    *I think the clustering of events is very interesting and well done. However, I think that one important parameter is missing. What was the event size of these events? Was event size different between the clusters and did therefore influence DOC export in different ways? I think that you could elaborate much more on the reasons for the occurrence of the different event types (see specific comment).*

100   We thank you for noticing the quality and the relevance of the clustering. If by event size you mean precipitation event size, unfortunately, we did not have this information for two individual events: events Aa and Ab. This information was added to the "Rainfall" section in material and methods.

      *Line 225: "Rainfall was measured from July 2018 to May 2020 using a tilting bucket rain gauge (Onset, 0.2 mm)."*

105   A mention was also added to the section describing the cluster in material and methods.

      *Line 315-316: "As precipitation data were not available for all events (i.e., Aa and Ab), precipitation-related variables were excluded from the clustering to keep the maximum number of events."*

      Also, it is because of the absence of precipitation data for the complete time series that we used
110   the term "flood event" rather than "storm event" along the manuscript since our analysis is based on WTD rather than rainfall. We invite the reviewer to look at Fig 3a to visualize the complete rainfall time series and the period when data is missing (before 31 July 2018).

      In this context, we would not exclude these events because of the absence of precipitation data while, on the other hand, we had variables and indices that could give a good understanding of
115   DOC export patterns during flood events.

      However, we consent that the discussion and particularly in section 5.3, we did not support our interpretation with precipitation events while it is a good indicator of what is happening in our catchment. Consequently, the discussion was adjusted to incorporate elements of discussion on precipitation.

120   *Line 545-547: "While the cluster 1 was characterized by a ΔWTD slightly higher than the average (Fig. 6b) and despite precipitation event 2 days before the flood (AP2) which was twice lower cluster 2 and more than three times lower than cluster 3, it also presented the lowest WTDinitial (-0.30 m; Table 3)."*

      *Line 556-559: "The high WTDinitial might indicate that these events succeeded a previously 'wet'*
125   *period which was confirmed by higher precipitation 14 days before the event (Table 3) compared to cluster 1 and similar to cluster 3 but also by a P-Q lag time (i.e., the lag time between the precipitation event and the increase of discharge in the stream) lower than other clusters (Table 3)."*

      *Line 566-569: "Event Bb, which is the only one with precipitation data, exhibit the highest*
130   *precipitation during the flood event but also the highest AP2, more than three times higher than cluster 3 and two times higher than cluster 2 (Table 3)."*

*Line 580-581: "The event Bb presented highest AP2 and total precipitation (Table 3) leading to an important ΔWTD (Fig. 6.b)."*

**You often write C (in units) but I think you mean DOC. Either use DOC or make clear in the beginning that C refers to DOC in your case.**

135

As you rightly mentioned, units of DOC exports should be in $g\ m^{-2}\ y^{-1}$. The unit was corrected in the manuscript and changes were reported in the *specific comments* section.

**Specific comments**

**L25 – Could you state at which interval you monitored the WTD?**

140    The measurement interval was added to the phrase.

*Line 26-27: "Hydrological variables, such as stream outlet discharge and the peatland water table depth (WTD), were continuously monitored at 1h intervals for 2 years."*

**L45 – Please insert „it" before „is crucial".**

"It" was added before "is crucial".

145    *Line 47-48: "In the context of a net ecosystem carbon budget, quantifying DOC exports, as well as particulate organic carbon (POC) and dissolved inorganic carbon (DIC) exports, it is crucial to evaluate how much carbon is lost through this pathway (Webb et al., 2019)."*

**L58 – Do you mean "Strong positive relationships"? Otherwise, one could think that there were strong relationships (which might be negative) and positive relationships (which might be**
150    **weak).**

The phase was corrected according to your comment.

*Line 60-62: "Strong positive relationships have already been established between the surface of a catchment covered by peat and the exported DOC to surface waters (Billett et al., 2006; Laudon et al., 2011; Olefeldt et al., 2013)."*

155    **L63 – Consider inserting "total" before "surface".**

We think that the "total" is implicit but was added to the phrase.

*Line 64-67: "Most of the previous studies have presented DOC exports normalized to the total surface of peatland-dominated catchments rather than normalized to the peatland surface area within the catchment (Köhler et al., 2008, 2009; Worrall et al., 2009; Dinsmore et al., 2013; Dick et*
160    *al., 2015), possibly leading to underestimating DOC exports."*

**L91-L92 I agree, but could you add references for this statement and give possible explanations? Also, some studies have shown that dry conditions could hinder DOC production (e.g. see references within Kalbitz et al. (2000))**

Pertinent references were added to support this statement.

165    *Line 94-95: "Previous studies have highlighted that those long periods between rainfall events favour DOC production (Clark et al., 2007; Glatzel et al., 2006; Grand-Clement et al., 2014)."*

**L195 At which depth were the wells installed?**

Wells were installed at two meters depth into the peat as it was now proceeding in the method section,

170    *Line 216-218: "Water table depth (WTD) was recorded hourly at the six wells (Fig. SI.2) inserted at about two meters depth into the peat and equipped with a water-level data logger (HOBO, Onset, USA) for continuous hourly measurement of WTD and temperature, from June to October 2018 and from June to October 2019 as described in Prijac et al. (2022)."*

**L122 – Do you have an idea about how this microtopography could influence the DOC dynamics**
175    **at your sites? Recent studies have shown that microtopography can be important for chemical and hydrological processes (Blaurock et al., 2022; Diamond et al., 2021; Mazzola et al., 2021).**

Parallels works conducted in our study site about CO2 and CH4 emissions, which will soon be submitted, will integrate the effect of microforms on those fluxes. However, our work did not integrate this variable into our study despite the fact that we are concerned that it also could play
180    a role in DOC dynamics. It is also due to the sampling design. Indeed, wells were installed from the top of the bog dome to the stream outlet rather than in different microforms. Then, we unfortunately do not have a satisfying resolution to explore this aspect of DOC dynamics.

**L133 This number doesn't seem to be correct.**

The correct number is 191.5 degree days above zero and was corrected into the manuscript.

185    *Line 142-143: "An average monthly positive temperature occurs from May to October with 191.5 growing degree days above zero (Havre-Saint-Pierre meteorological station, mean 1990–2019, Environment of Canada)."*

**L137 This is not really the event size but rather the daily precipitation. But do you have event size data as well? This would be interesting as daily precipitation only gives us an average.**

190    Daily precipitation is here to describe meteorological data for the studied site. The precipitation event was more effectively used in following sections of the manuscript, and also was more used in the manuscript in order to support interpretation about the clustering as you can see in general comments of our response (line 90-121 of the present document).

**L175 How many samples did you use to calibrate? In Figure 3, it looks like you took 6 samples,**
195    **which would be a very low number for a calibration. Do you have the calibration curves and R2 values? You could maybe add them to the supplementary material.**

The number of samples taken for fDOM calibration (n = 69) was added to the manuscript.

*Line 176: "During the 2018 and 2019 growing seasons, punctual water samplings were taken in the stream (n = 69)."*

200  In addition, calibration curves were added to the supplementary information (Table SI.1) and the table was referred in the method section of the manuscript.

*Line 185-187: "The fDOM measurements were used to determine DOC, considering the relationship f(fDOM) = [DOC], where fDOM is the corrected signal fluorescence of DOM measured in quinine sulfate units (QSU) and [DOC] is the dissolved organic carbon concentration in mg C L-1*
205  *(Table SI.1)."*

**L192 Which uncertainties do you mean? Can you specify?**

The necessity to use modelled discharge values during the spring thaws was caused by 1) the damaging of SR-50A distance sensor during the 2019 spring thaw and 2) by the impossibility to measuring discharge during the spring thaw because the water level in the stream exceeds the "V-
210  shaped" weir and the stream got out of its bed. The section was clarified in the manuscript.

*Line 209-214: "Discharge monitored data during the spring thaw was not available due to the absence of monitored data from the ultrasonic distance sensor SR-50A during 2019 spring freshet, because the sensor was damaged during the flood and because of measurements during 2020 spring thaws cannot be measured as the flooded section exceed the stream bed and the Thomson's*
215  *triangular notch equation cannot be applied. Consequently, daily water discharge was modelled during the whole studied period, using the Peatland Hydrologic Impact Model (PHIM) developed by Guertin et al. (1987) and detailed by Riahi (2021)."*

**L206 Did you also measure snowfall? Was snowfall counted as precipitation?**

Unfortunately, Snowfall was not measured in our site during the study period. However,
220  precipitation measurements at Havre-Saint-Pierre airport mentioned that, in average, snowfall account for 58% of the annual precipitation (line 133).

**L220 The 10th quantile of which period?**

As the methodology was slightly modified, this phrase was removed.

**L290-293 Do you need all the decimals here? The error margin is probably much larger.**

225  We kept three decimals along the manuscript for discharge values presented in m3 s-1.

*Line 328-331: "The average annual Q was 0.020 m3 s-1 in 2018–2019 and 0.017 m3 s-1 in 2019–2020. During the growing season, the lowest monthly average discharge occurred in July of each year, with 0.010 m3 s-1 in 2018–2018 and 0.007 m3 s-1 in 2019–2020. In 2018–2019, the highest discharge was 0.068 m3 s-1 measured in June 2018 and in 2019–2020 it was 0.100 m3 s-1*
230  *measured in September 2019"*

*L315-L316 Maybe I missed it, but I think that you do not further elaborate on the importance on porewater temperature for DOC stream concentrations. Does the porewater temperature add information to stream temperature? What could be reasons for the negative correlation? I wonder why this is brought up here quite prominently but then not used in the discussion.*

235     The relation between peat porewater temperature was mentioned here as its importance emerged in the random forest. However, this information is constrained to the above mentioned application (i.e., Random Forest model) and did not give better information than hydrological variables (WTD and Q) when considered/plotted individually against DOC concentrations. Consequently, this phrase was removed from the manuscript.

240     *L323-324 I think it would be okay if you mention the rounded values again.*

The catchment and peatland surface area presented here corresponds to the values used to calculate specific DOC exports. Given the importance of these values for our study, we think it is meaningful to keep them not rounded in the manuscript.

*L401-402 With accretion you mean an increase of DOC concentrations in the stream? I am not*
245     *sure if accretion is the right word here? Maybe accumulation? Or maybe add "in the stream".*

As you rightly suggest, we added "in the stream" in the phrase.

*Line 443-444: "The positive FI and β index (Table 3 and Fig. 5) indicate accretion of DOC in the stream during flood episodes and reveal a transport limitation of DOC (Vaughan et al., 2017; Zarnetske et al., 2018)."*

250     *L432 These numbers refer to DOC only. If you use C, this would include DIC and POC in my point of view. As your write later, DOC only accounts for a small percentage of total C exports.*

For all numbers presenting DOC exports, the units are changed from g C m-2 y-1 to g DOC-C m-2 y-1.

*Line 494-495: "The annual exports using this approach were 1.9 g DOC-C m-2 y-1 in 2018–2019*
255     *and 1.3 g DOC-C m-2 y-1 in 2019–2020."*

*Line 506-507: "In this study, DOC exports are lower than those previously measured in undisturbed boreal peatland drainage streams, which varied from 3.7 to 18.0 g DOC-C m-2 y-1 (Köhler et al., 2008, 2009; Juutinen et al., 2013; Leach et al., 2016)."*

*Line 512-514: "However, even in a scenario of spring freshet contributing to 50% of DOC exports,*
260     *estimated annual DOC exports would be about 2.2 and 1.6 g DOC-C m-2 y-1 for 2018-2019 and 2019-2020 respectively, remained in the lower range of those measured in the literature (3.7-18.0 g DOC-C m-2 y-1)."*

Changes were also made in the *material and methods* section.

*Line 249: "The DOC load at the outlet of the catchment (g DOC m-2 year-1) was calculated as in equation (1)."*

As well as in the *results* section.

*Line 365-375: "The specific annual DOC exports were 1.87 g DOC m-2 y-1 for June 2018–May 2019 and 1.27 g DOC m-2 y-1 for June 2019–May 2020 (Table 2 and Fig. 4). The strategy used to calculate the specific DOC exports by distinguishing high flow and low flow provides a better estimation of exports. If the most conservative surface (i.e., the catchment area) would have been used to calculate the specific exports, it would have been 1.46 g DOC m-2 y-1 in 2018–2019 and 0.99 g DOC m-2 y-1 in 2019–2020.*

*This approach provides a range for the plausible specific DOC exports from the peatland between 1.46 and 1.91 g DOC m-2 y-1 for 2018–2019 and between 0.99 and 1.29 g DOC m-2 y-1 for 2019–2020."*

**L467 ff It would be really interesting to know the different event sizes of the clusters. Do you have data on this? Event size could significantly influence DOC export.**

Information about precipitation during these flood events were added to the manuscript, according to your comments. It was more detailed in general comments section (line 91-212 of the present document).

*Line 545-547: "While the cluster 1 was characterized by a ΔWTD slightly higher than the average (Fig. 6b) and despite precipitation event 2 days before the flood (AP2) which was twice lower cluster 2 and more than three times lower than cluster 3, it also presented the lowest WTDinitial (-0.30 m; Table 3)."*

**L482 Again, better use DOC.**

As mentioned previously, changes were made along the manuscript.

**L483-L484 Maybe write "less negative". At first, I thought "high" meant a large magnitude of the HI, which got me confused about your interpretation.**

"High" was replaced by "less negative" in the manuscript.

*Line 563-565: "Although the threshold of the lateral discharge generation was easily exceeded, the less negative HI suggests that DOC was mostly exported from sources close to the stream (Tunaley et al., 2017)."*

**L497 "a single event"**

The manuscript was corrected according to your comment.

*Line. 581-582: "These data suggest that the magnitude of a single event is at least as important as several events (Raymond & Saiers, 2010)."*

**L498-505 This is really interesting and I think you could elaborate much more on the different mechanisms which lead to the high DOC export. For example, the longer dry period could lead to an accumulation of DOC, which is being produced but not exported (Bb). And why is the Aa event so important? Is snowmelt the reason?**

The paragraph on this point was completed and improved according to the reviewer's comment. We now stress the importance of interannual meteorological conditions on DOC exports during the spring freshet. Nevertheless, the limited period covered by our study did not allow us to interpret more precisely those periods.

*Line 584-590 : "The Aa event occurred at the end of the spring freshet, which is known as an important period of DOC exports (Tiwari et al., 2018). However, similar events were not observed during 2019 snowmelt and event Ba that occurred during this period was attributed to cluster 2 (Fig. 6.a). However, similar amounts of DOC were exports during May 2019 compared to June 2018 that could reveal a delayed spring thaw in 2019 compared to 2018. Previous studies observed that variability in DOC exports can be influenced by interannual variation of meteorological conditions (Ågren et al., 2010; Dinsmore et al., 2013; Tiwari et al., 2018). The period covered by our study limits this type of interpretation but it is reinforcing the necessity of long-term DOC exports monitoring (Webb et al., 2019)."*

Concerning the case of the event Bb, we think the point raised by the reviewer was already quite well described in the discussion. Yet, the phrase was completed to be more precise.

*Line 593-595: "This may coincide with conditions that have previously been described as favourable for DOC production which is accumulated within the peat during dry periods (Clark et al., 2007, 2009; Dinsmore et al., 2013)."*

**L504 "initiated"**

The manuscript was corrected accordingly.

*Line 595-596: "Then, the large rainfall events occurring before the event initiated an important WTD increase that leads to DOC mobilization (Table SI.3; Grand-Clement et al., 2014; Zhu et al., 2022)."*

**Figure 2 Add to the caption that you mean the DOC flux in the stream.**

The caption of the figure was changed in consequence.

**Figure 2.** (a) Relation between hourly measurements of the water table depth (WTD, in m) and stream discharge ($Q$, in $m^3$ $s^{-1}$). The colour represents the day of the year and the dashed line corresponds to the logarithmic relation between WTD and $Q$. (b) Relation between the hourly measurements of WTD (m) and hourly DOC flux *in the stream* (g DOC-C $h^{-1}$). The colour represents the hydrological state according to the hidden Markov model and the dashed line corresponds to the logarithmic relation between WTD and DOC flux.

**Figure 3 In the caption b) is missing but d) is double.**

The caption of the figure was changed in consequence.

**Figure 3.** Times series of (a) stream and porewater temperature and precipitations, *(b)* water table depth (WTD), *(c)* log-transformed stream discharge (log*Q*), (d) the dissolved organic carbon (DOC) concentration in the stream and e) DOC exports, from June 2018 to May 2020. Colours in the (b)–(e) correspond to the periods of flood (in blue) and low flow (in red). Grey vertical bars represent individual storm events. Yellow diamonds represent DOC concentration analyses from punctual sampling at the stream outlet.

***Figure 4 You could add titles above the panels showing the corresponding year.***

The figure was changed in consequence.

[Figure]

***Table 2 Check the superscription of units in Table 2b).***

Units were checked and homogenized between both table 2.a and 2.b.

(a)

| Month | 2018–2019 DOC flux (g DOC-C $m^{-2}$ $month^{-1}$) | | 2019–2020 DOC flux (g DOC-C $m^{-2}$ $month^{-1}$) | |
|---|---|---|---|---|
| | High flow | Low flow | High flow | Low flow |
| June | 0.452 | 0.000 | 0.102 | 0.008 |
| July | 0.130 | 0.022 | 0.000 | 0.009 |
| August | 0.167 | 0.053 | 0.229 | 0.016 |
| September | 0.144 | 0.011 | 0.327 | 0.012 |
| October | 0.208 | 0.003 | 0.080 | 0.005 |
| November | 0.208 | 0.003 | 0.099 | 0.000 |
| December | 0.000 | 0.010 | 0.060 | 0.001 |
| January | 0.000 | 0.003 | 0.000 | 0.010 |
| February | 0.000 | 0.004 | 0.000 | 0.008 |
| March | 0.000 | 0.006 | 0.000 | 0.010 |
| April | 0.052 | 0.008 | 0.136 | 0.001 |
| May | 0.418 | 0.000 | 0.157 | 0.000 |
| Total per conditions | 1.727 ± 0.72 | 0.138 ± 0.099 | 1.189 ± 0.551 | 0.079 ± 0.045 |
| Specific flux | 1.865 ± 0.746 | | 1.268 ± 0.348 | |

(b)

| | 2018–2019 | | | 2019–2020 | | |
|---|---|---|---|---|---|---|
| | Proportion of measurements (%) | Flux (g DOC-C $m^{-2}$ $y^{-1}$) | Proportion of flux (%) | Proportion of measurements (%) | Flux (g DOC-C $m^{-2}$ $y^{-1}$) | Proportion of flux (%) |
| High flow | 59.1 | 1.727 | 92.6 | 44.1 | 1.189 | 93.8 |
| Low flow | 40.9 | 0.138 | 7.4 | 55.9 | 0.079 | 6.2 |
| Total | 100.0 | 1.865 | 100.0 | 100.0 | 1.268 | 100.0 |

345

*Figure 5 I understand that you used normalized values here to better compare the hysteresis patterns. However, like this information on event characteristics gets lost. I wonder if you could prepare the same figure with unnormalized data for the supplementary material? Also, is the count always hourly? Add this information to the caption.*

350 For the "exercise" we did the figure of hysteresis patterns with the non-normalized data. However, we did not consider adding it in the manuscript nor in supplementary materials. As you can see, it did not provide more pertinent information compared to the figure 5. In addition, information about event characteristics (particularly concerning minimum, maximum and increase of Q and DOC) can be found elsewhere in the manuscript (e.g., in the clustering (Fig. 6) or in table 3 and
355 SI.3).

[Figure]

The caption of the figure was changed in consequence.

**Figure 5.** The hysteretic relations between *hourly measurements* of normalized stream discharge ($Q$) and normalized dissolved organic carbon (DOC) for the events of (a) cluster 1, (b) cluster 2 and (c) cluster 3. The colour represents the count of the measure, from 0 at the beginning of the event to the end. The hysteresis index (HI), the flushing index (FI) and the $\beta$ index are presented for each event.

References

Blaurock, K., Garthen, P., Da Silva, M. P., Beudert, B., Gilfedder, B. S., & Fleckenstein, J. H., et al. (2022). Riparian Microtopography Affects Event-Driven Stream DOC Concentrations and DOM Quality in a Forested Headwater Catchment. *Journal of Geophysical Research: Biogeosciences*, *127*(12). https://doi.org/10.1029/2022JG006831

Diamond, J. S., Epstein, J. M., Cohen, M. J., McLaughlin, D. L., Hsueh, Y.-H., Keim, R. F., & Duberstein, J. A. (2021). A little relief: Ecological functions and autogenesis of wetland microtopography. *WIREs Water*, *8*(1). https://doi.org/10.1002/wat2.1493

Kalbitz, K., Solinger, S., Park, J.-H., Michalzik, B., & Matzner, E. (2000). Controls on the dynamics of dissolved organic matter in soils: A review. *Soil Science*, *165*(4), 277–304. https://doi.org/10.1097/00010694-200004000-00001

Mazzola, V., Perks, M. P., Smith, J., Yeluripati, J., & Xenakis, G. (2021). Seasonal patterns of greenhouse gas emissions from a forest-to-bog restored sites in northern Scotland: Influence of microtopography and vegetation on carbon dioxide and methane dynamics. *European Journal of Soil Science*, *72*(3), 1332–1353. https://doi.org/10.1111/ejss.13050

375

---

## Author Comment (AC2)

**Response letter to the reviewers: Manuscript hess-2022-426**

In this response letter, the reviewer's comments are in **italic bold black**, our responses are in blue and significant new text added to the manuscript are in *italic green*. Changes made in the manuscript are tracked and line number referred to the revised manuscript.

**Reviewer #2**

*The authors conducted a study to examine the relationship between hydrological connectivity and carbon exports in a peatland-dominated watershed. They aimed to achieve several objectives: a) establish the connection between dissolved organic carbon (DOC) exports and peatland hydrology, b) quantify the lateral export of DOC into the stream at the catchment scale, and c) identify patterns of DOC mobilization during high-flow events. They propose a method to estimate carbon exports based on the relative contribution of the peatland in relation to the whole watershed.*

*The authors deployed a multiparameter probe at the watershed outlet to measure fDOM, turbidity, DO, SpC, water temperature, and pH hourly from June 2018 to May 2020. The relationship between fDOM and DOC was assessed by analyzing temperature-corrected fDOM signals and DOC concentrations obtained from grab samples collected during 5 and 4 sampling events in 2018 and 2019, respectively. Streamflow was estimated from year-round water level measurements, with the relationship calibrated using field streamflow measurements, except for the spring thaw period, where a PHIM model was used. Hidden Markov chains were used to classify the streamflow data into high and low flow periods. Water table elevations were recorded hourly at six wells during the growing season from June 2018 to October 2020.*

*The design of the study is sound. The paper is well-written, the figures are clear, and the methods section includes good detail. Altogether, the study is a good contribution to the field and improves our understanding of the role of boreal headwater streams in the carbon cycle. I have some recommendations I think will improve the flow of the manuscript and its impact. My major recommendation is to tone down some of the statements regarding WTD and Q relationships since the study does not technically prove lateral flow directionality (see comments below regarding Lines 399 and 515) and to slightly expand the discussion to explain better the results on the context of other studies done in the same site and other boreal streams (see comment below regarding L457).*

We thank the reviewers for their comments, and we are pleased that you recognize the overall quality of our manuscript. We are concerned by the recommendations you made and we are convinced that they helped us to improve the quality of the manuscript. You will find responses and changes we made to the manuscript according to your comments.

*Specific comments:*

*L190 "The calculation method was…" replace by "The calculation method for…. was calculated…."*

The phrase was changed according to your suggestion and the calculation method of the discharge were precised.

40    *Line 199-201: "The calculation method for the discharge was described by Taillardat et al. (2022). The distance between the surface water and the ultrasonic distance sensor gives the flooded vertical area in the 'V-shaped' weir and the Thomson's triangular-notch equation allow calculating the discharge from water-level measurements (Shein, 1979)."*

**L191 Please explain how you measured streamflow used in your calibration.**

45    This section of the methodology was rewritten to clarify the method according to this comment and the previous one. A first ultrasonic sensor was installed during the 2018 growing season (see previous comment). But as it was damaged during the 2019 spring freshet, it was replaced by a water-level data logger starting from June 2019.

*Line 202-208: "Starting from June 2019, a water-level logger (U201-04, Hobo, Onset, USA) was*
50    *installed at the stream outlet to replace the ultrasonic distance sensor, damaged during the spring freshet. Water-level-discharge rating curves were calculated following the method described by (Taillardat et al., 2022). Discharge was measured at the stream outlet using a portable flow velocity probe (Flo-mate model 2000, Marsh-McBirney Inc., USA) measuring water velocity in a stream cross-section at subsections of 20 cm with intervals. The cumulative discharge (Q; in m3 s-*
55    *1) was measured by summing the discharge obtain for each subsection by Equation (1) where V is the water velocity measured by portable flow velocity probe (in m s-1) and A is the flooded vertical area (in m2) and obtain by multiplied depth (in m) to the width of the section (in m)."*

**L192-193 "daily water discharge was modelled using PHIM" – please clarify if you modelled streamflow only during spring thaws or for all year round.**

60    The modelled period was clarified. Also, we precise reasons why PHIM model was used.

*Line 209-214: "Discharge monitored data during the spring thaw was not available due to the absence of monitored data from the ultrasonic distance sensor SR-50A during 2019 spring freshet, because the sensor was damaged during the flood and because of measurements during 2020 spring thaws cannot be measured as the flooded section exceed the stream bed and the Thomson's*
65    *triangular notch equation cannot be applied. Consequently, daily water discharge was modelled during the whole studied period, using the Peatland Hydrologic Impact Model (PHIM) developed by Guertin et al. (1987) and detailed by Riahi (2021)."*

**L195 - How deep were the wells?**

Wells were installed at 2 meters depth into the peat as it was now clarified in the method section,

70    *Line 216-218: "Water table depth (WTD) was recorded hourly at the six wells (Fig. SI.3) inserted at about two meters depth into the peat and equipped with a water-level data logger (HOBO, Onset, USA) [...]."*

**L196 – "water-level data logger… from June 2018 to October 2020" – is that right? Or were they only deployed during the growing season?**

75 As it was presented in Fig.3.b, water-level data logger monitored WTD from June to October 2018 and from June to October 2019 only. Harsh climatic conditions have emptied WL data loggers batteries and as we cannot collect data outside of the growing season, WTD were only available for these periods. The manuscript was adjusted.

*Line 216-218: "[...] for continuous hourly measurement of WTD and temperature, from June to*
80 *October 2018 and from June to October 2019 as described in Prijac et al. (2022)."*

**L203 – please include at how many sites.**

The phrase was removed (line 215) as those data were not used afterwards. Is the water table temperature measured with water level logger which was used.

**L220 – "Gap filling (..) could not be performed during (…) the non-growing season due to the**
85 **bad quality of the model (i.e., low linear relationships between the predicted and measured values" however, in L146-147 you indicate that grab samples were only collected during the growing season. Which one is correct?**

There was an error in the previsouly submitted version of the manuscript. Indeed, the gap filling correctly performed for the complete daily time series (from June 2018 to May 2020) which
90 allowed us to model daily DOC concentrations during this period (i.e., non-growing season). The last version of the random forest method description was included in the manuscript. We apologies for the confusion.

*Line 240-246: "Gap filling of the DOC concentration was also performed during the rest of the time series (i.e., non-growing season). The same method was applied on the daily-interval data set to*
95 *model the missing DOC concentrations (51.3% of the data set). The data set contained the PHIM simulated discharge, water temperature, pH, dissolved oxygen saturation and specific conductivity. The training data set corresponded to 26% of the data set and validation data set corresponded to 22.7% of the data set. The validation test of the random forest model gave a relatively good fit with a strong positive correlation between observed and modelled DOC*
100 *concentration (cor = 0.84; p-value < 0.0001), the mean root-square residuals was 2.15 and the percentage of variance explained by the model was 71% (p-value < 0.0001; Fig. SI.1.b)."*

**L222 – "the 10th quantile of the DOC concentration was used to fill the gaps": Please explain the rationale behind using the 10th quantile.**

This phrase was removed as the gap filling method was rightly applied to the daily-interval dataset.

105 **L232 – "HMM was used to classify the time series": Specify which time series – flow with PHIM outputs? Or original water level?**

We specify in the manuscript that HMM was used in both daily and hourly interval datasets.

*Line 270-271: "The HMM was applied on both 1h-interval data set and on PHIM modelized daily-interval datasets."*

110 *L244 "For each flow event" – Replace by "For each of the 12 flood events"*

The manuscript was corrected following your suggestions.

*Line 280: "For each of the 12 flood events, several descriptive and quantitative indicators were calculated; they are described in Table 1."*

*Figure 3 – add units to discharge*

115 The unit was added to the panel (c) of figure 3.

[Figure]

*L373 – "Although the events of cluster 1 had the highest ΔDOC" but in L369 "but the events in cluster 1 presented the lowest ΔDOC".*

There was typo here as the highest ΔDOC was measured for cluster 3 rather than for cluster 1. The
120 manuscript was corrected.

*Line 414-416: "Although the events of cluster 3 had the highest ΔDOC, the events of cluster 2 had the highest Qmax and ΔQ, namely 0.086 ± 0.018 and 0.065 ± 0.022 m3 s-1, respectively."*

*L399 – "The increase in WTD led to an increase in Q" – here and in general, I would tone it down, maybe "the increase in WTD coincided with an increase in Q". I don't believe you are strictly*
125 *proving causation.*

It is a good point you mentioned here. Even if we agree to soften this statement, in a peatland-dominated watershed, we could reasonably hypothesize that the peatland significantly contributes to the stream discharge through subsurface runoff, following the increase of the WTD (Bishop et al., 2004; Tunaley et al., 2016, 2017).

*Line 440-441: "The increase in WTD coincide to an increase in Q and DOC concentrations at the outlet and, consequently, to an increase in DOC exports (Fig. 2)."*

**L457-459 – "DOC only accounts for 13.6%-18.8% of the total aquatic carbon" – it would be worth expanding this paragraph since it can provide a comprehensive context for your research, for example, how did you account for total carbon – is that including particulate exports? Can you use the data from Taillardat et al 2022, your data, and bibliography to provide evidence that the lower-than-expected DOC exports are due to higher rates of transformation to GHG?**

Figures (13.6-18.8%) were not good as the wrong GHG flux was considered. The GHG flux is 1.08 g m$^{-2}$ y$^{-1}$ (Taillardat et al., 2022) and DOC fluxes were 1.87 and 1.27 g m$^{-2}$ y$^{-1}$ for 2018-2019 and 2019-2020 respectively. Given that change, the proportion of DOC exports is coherent with data from the literature that ranged this contribution between 46 and 95% (Roulet et al., 2007; Nilsson et al., 2008; Worrall et al., 2008; Dyson et al., 2011; Holden et al., 2012; Huotari et al., 2013; Dinsmore et al., 2013; Leach et al., 2016). The paragraph was modified according to this change. Also, it was precise in these changes that aquatic exports only include GHG (CO2+CH4) and DOC.

*Line 525-530: "In terms of total carbon flux in our studied peatland, Taillardat et al. (2022) estimated the stream carbon GHG (CO2 and CH4) aquatic exports as 1.08 g GHG-C m-2 y-1. It gives a total aquatic carbon exports (GHG + DOC) ranged between 2.35 and 2.95 g C m-2 y-1 and a contribution of DOC exports accounting for 54-63% of the total aquatic carbon exports. This is in line with previous studies which observed a DOC contribution to aquatic carbon flux ranged between 46 and 95% (Roulet et al., 2007; Nilsson et al., 2008; Worrall et al., 2008; Dyson et al., 2011; Holden et al., 2012; Huotari et al., 2013; Dinsmore et al., 2013; Leach et al., 2016)."*

**L515 – "given the lack of a direct link between peat porewater discharge and DOC exports from the stream" – Maybe I missed that detail, but my understanding is that you did not measure WTD during the non-growing season, when the majority of low flow occurs. If that is the case, I would tone down the statement and acknowledge the limitations of the study in this regard.**

You raised an interesting point, then we precise "during the growing season" in the phrase to avoid any confusion. However, if we agree that the absence of WTD measurements during some low-flow periods could be a limitation, it is important to consider that those periods also correspond to the ice-covered period. In this context, we hypothesized that DOC exports are not only limited by the hydrological connectivity but also by the stream freeze that leads to very low discharge (Fig. 3.c).

*Line 605-607: "By contrast, during the low-flow periods, the catchment area is considered the conservative surface reference in the calculation given the lack of a direct link between peat porewater discharge and DOC exports from the stream observed during the growing season."*